# Mitigating Reward Hacking in Multi-Reward Optimization for Text-to-Image Generation

## Abstract

Text-to-image generation models have achieved remarkable progress in preference optimization, yet achieving robust alignment across diverse reward models remains a significant challenge. Existing multi-reward fusion approaches rely on weighted summation, which is costly to tune and insufficient for balancing conflicting objectives. More critically, optimization with reward models is highly susceptible to *reward hacking*. We theoretically suggest that using a unified global upper bound as the optimization target may induce *reward hacking* in certain samples. In addition, optimization with weak reward models is particularly prone to exacerbating this risk. To address this issue, we propose a Pareto frontier-guided optimal transport framework, which constructs a frontier for each prompt as the optimization target and maps generated samples within the same batch to their corresponding frontiers. Based on the characteristics of the reward models, we further design both online and offline optimization strategies to adapt to the distinct requirements of different reward models. Finally, we introduce the Joint Domination Rate (JDR) and Joint Collapse Rate (JCR) as more principled metrics for evaluating multi-reward optimization. Experimental results demonstrate that, compared with strong baselines, our method achieves a 10% performance improvement, effectively mitigating reward hacking while enhancing multi-reward alignment.

## 1 Introduction

With the rapid advancement of text-to-image generation Sohl-Dickstein et al. (2015); Song & Ermon (2020); Rombach et al. (2022); Podell et al. (2023); Esser et al. (2024), numerous post-training optimization methods have emerged Ziegler et al. (2020); Stiennon et al. (2022); Black et al. (2024); Dhariwal & Nichol (2021); Li et al. (2024b); Rafailov et al. (2024); Fan et al. (2023), among which aligning model outputs with human preferences via reward models has become a key research direction. Early approaches typically relied on a single human-preference reward model Xu et al. (2023); Zhang et al. (2024). While simple to implement, these methods face two critical issues: first, their optimization objective is overly narrow; second, they are highly susceptible to *reward hacking*, where reward values continue to increase while the visual quality of generated images deteriorates.

To mitigate this issue, subsequent research proposed multi-reward joint optimization Eyring et al. (2024); Agarwal & Aggarwal (2023); Wei et al. (2024); Deng et al. (2024); de Langis et al. (2024); Lee et al. (2025), which fuses multiple reward signals to provide richer and more constrained guidance. Although the combined constraints of multiple rewards can partially suppress reward hacking, such methods introduce significant and costly weight-tuning requirements and still fail to address the fundamental problem. We argue that both single-reward optimization and multi-reward joint optimization share the same underlying flaw: different prompts span image domains with heterogeneous reward ranges and upper bounds, yet existing methods commonly ignore this heterogeneity and adopt a unified global optimization target. Even with sophisticated weight adjustments that alter the relative strength of different rewards, the underlying global bound remains unchanged and cannot fundamentally prevent reward hacking. We theoretically prove that, under heterogeneous reward upper-bound distributions, using a unified global target inevitably induces reward hacking for certain samples, regardless of whether single- or multi-reward optimization is applied.

To address this root cause, we propose a **Pareto-frontier-guided optimal transport optimization algorithm**. The core idea is to explicitly respect the heterogeneous reward boundaries of different

image domains rather than blindly applying a global target. Specifically, for each prompt we construct a dedicated Pareto frontier as the optimization target, treat the generated samples within the same prompt batch as the source distribution, and use optimal transport theory to map them to the corresponding frontier as the target distribution. Furthermore, based on the characteristics of the adopted reward models, we design two complementary optimization strategies: for strong reward models, we adopt an *online strategy*, dynamically collecting and updating Pareto frontiers during training to explore superior solutions; for weaker reward models prone to collapse, we adopt an *offline strategy*, precomputing large-scale samples and extracting their Pareto frontiers to provide stable optimization guidance.

Finally, to address the lack of precise evaluation metrics for multi-reward optimization, we introduce two new indicators: **Joint Domination Rate (JDR)** and **Joint Collapse Rate (JCR)**. Existing evaluation methods typically measure overall performance by averaging reward improvements across objectives, but such averages may not reflect consistent improvements across all rewards for individual samples, leading to inaccurate assessments. JDR measures the proportion of samples that simultaneously outperform the baseline across all reward functions, while JCR measures the proportion of samples that degrade across all rewards. Compared with traditional mean-based metrics, these indicators provide a more faithful assessment of synergistic gains in multi-reward optimization.

Our main contributions are as follows:

1. We theoretically show that a unified global target under heterogeneous reward bounds inevitably causes reward hacking in both single- and multi-reward settings.
2. We propose a Pareto-frontier-guided optimal transport approach with prompt-specific frontiers and online/offline strategies tailored to reward model properties.
3. We introduce Joint Domination Rate (JDR) and Joint Collapse Rate (JCR) for more accurate evaluation of multi-reward optimization.

## 2 MECHANISMS BEHIND REWARD HACKING

When using reward models to enhance performance in text-to-image generation, previous optimization methods Clark et al. (2024); Eyring et al. (2024) employ a global constant subtracted from the reward function in the loss function to maximize rewards. However, these approaches suffer from reward hacking. While early stopping strategies mitigate this issue by halting training before collapse occurs, they failed to address the underlying causes of reward hacking. In this work, we present a causal analysis of reward hacking and propose methods to verify our hypothesis.

**Problem Setup.** Consider a text-to-image (T2I) framework with prompt set $\mathcal{P} = \{p_1, \ldots, p_n\}$, where each prompt $p_i$ contributes to a distinct image domain $\mathcal{D}_i \subseteq \mathcal{X}$, and $\mathcal{X}$ denotes the overall image space. A reward function $R : \mathcal{X} \to \mathbb{R}$ evaluates generated images, and $Q : \mathcal{X} \to \mathbb{R}$ denotes human-perceived quality, reflecting comprehensive subjective human evaluation of images. For each domain $\mathcal{D}_i$, a *perceptually admissible* set $\mathcal{F}_i \subseteq \mathcal{D}_i$ is defined as the subset of samples for which the rewards positively correlated with human-perceived quality, *i.e.*, higher reward values correspond to improvements in human-perceived quality.

**Definition 1** (Reward Hacking). *Given prompt $p_i$ with its corresponding image domain $\mathcal{D}_i$ and $K$ reward functions $(R^1, R^2, \ldots, R^K)$, let $\overline{R_i^k} = \max_{x \in \mathcal{F}_i} R^k(x)$ be the $k$-th reward upper bound over the perceptually admissible feasible set $\mathcal{F}_i$, and let $\underline{Q_i} = \min_{x \in \mathcal{F}_i} Q(x)$ be the minimum human-perceived quality within $\mathcal{F}_i$. We define the set of* reward-hacked *images as:*

$$\mathcal{H}_i := \left\{ x \in \mathcal{D}_i \,\middle|\, \exists k : R^k(x) > \overline{R_i^k} \ \wedge \ Q(x) < \underline{Q_i} \right\}. \tag{1}$$

That is, samples with **at least one** reward dimension exceeding its feasible reward upper bound, yet with human-perceived quality below the feasible minimum.

### 2.1 REWARD HACKING BY NEGLECTING HETEROGENEOUS REWARD BOUNDS

**Reward Optimization Objective under Neglected Reward Bounds.** Direct optimization approaches Xu et al. (2023); Clark et al. (2024); Eyring et al. (2024) for T2I models using reward

signals typically reformulate reward maximization into a minimization problem via a surrogate loss:

$$\mathcal{L}(x) = C - \sum_{k=1}^{K} w_k R^k(x), \quad w_k \geq 0, \quad \sum_{k=1}^{K} w_k = 1. \tag{2}$$

Where $C$ is a constant upper bound on the aggregated reward. In practice, $C$ is defined as the weighted sum of the individual reward upper bounds, each computed as the supremum of $R^k(x)$ over all prompts $p_i$ within their feasible domains $\mathcal{F}_i$:

$$C = \sum_{k=1}^{K} w_k \sup_{i,\, x \in \mathcal{F}_i} R_i^k(x). \tag{3}$$

And the single-reward setting reduces to the special case $K = 1$. However, using a global upper bound $C$ creates a fundamental mismatch with real-world scenarios where different prompts have inherently different maximum achievable rewards (Appendix F). Each reward dimension not only varies in scale but also in difficulty across prompts, creating a complex heterogeneous optimization space. By enforcing a global reward bound, it ignores the prompt-specific feasible maxima, leading to biased gradient signals that can lead to reward hacking.

**Property 1** (Heterogeneous Reward Upper Bounds within Admissible Sets). *Each prompt $p_i$ induces a perceptually admissible domain $\mathcal{F}_i \subseteq \mathcal{D}_i$ associated with a distinct upper bound on the reward function. This can be formalized as the Heterogeneous Reward Upper Bounds property:*

$$\exists\, p_i \neq p_j \quad s.t. \quad \overline{R_i} := \sup_{x \in \mathcal{F}_i} R(x) \;\neq\; \overline{R_j} := \sup_{x \in \mathcal{F}_j} R(x), \tag{4}$$

*where $\overline{R_i}$ denotes the upper-bound within $\mathcal{F}_i$.*

This heterogeneity arises from prompt-dependent semantic constraints, biases in the reward model, and related factors. Consequently, no single global reward bound applies uniformly across all prompt-induced domains. Empirical visualizations further support this property, revealing distinct reward bounds across different prompts.(see Appendix F).

**Proposition 1** (Global Bound Induces Reward Hacking). *For any image domain $\mathcal{D}_i$ induced by a prompt $p_i$, let $\overline{S_i} = \max_{x \in \mathcal{F}_i} \sum_{k=1}^{K} w_k R^k(x)$ denote the maximal weighted reward attainable within its feasible set $\mathcal{F}_i$. By Property 1, domains exhibit heterogeneous reward upper bounds. Since the global constant $C$ is derived from a weighted combination of global reward upper bounds across all domains, there exist domains for which $C > \overline{S_i}$. When optimizing $\mathcal{L}(x) = C - \sum_{k=1}^{K} w_k R^k(x)$, the objective drives $\sum_{k=1}^{K} w_k R^k(x)$ beyond $\overline{S_i}$, typically by disproportionately increasing rewards in some dimensions. This results in $R^k(x) > R_i^{k*}$ for some dimension $k$, while simultaneously $Q(x) < \underline{Q_i}$ occurs. By Definition 1, this implies $x \in \mathcal{H}_i$. Additional proofs are given in Appendix B.*

## 2.2 REWARD HACKING VIA WEAK REWARD MODELS

To understand reward hacking in multi-reward optimization, we categorize existing reward models into strong and weak types based on how well their preference predictions align with actual human-perceived quality. In Appendix D.5, we preliminarily distinguish strong and weak reward models by evaluating their preference accuracy on high-quality human-preference datasets.

**Definition 2** (Strong Reward Model). *A reward function $R$ is categorized as a* strong reward model *$\mathcal{R}^S$ if and only if, for any prompt $p_i$ with its perceptually admissible feasible set $\mathcal{F}_i$, the correlation between $R$ and human-perceived quality $Q$ is bounded below by a positive threshold:*

$$R \in \mathcal{R}^S \iff \forall i,\, \forall x \in \mathcal{F}_i, \quad \mathrm{Corr}\big(R(x), Q(x)\big) \geq \tau,$$

*where $\tau > 0$ is a correlation threshold and $\mathrm{Corr}(\cdot, \cdot)$ denotes the Pearson correlation coefficient.*

**Definition 3** (Weak Reward Model). *A reward function $R$ is called a* weak reward model *$\mathcal{R}^W$ if, for some prompt $p_i$ with its perceptually admissible feasible set $\mathcal{F}_i$, the correlation between $R$ and human-perceived quality $Q$ is insufficient to provide reliable preference predictions:*

$$R \in \mathcal{R}^W \iff \exists i,\, \exists x \in \mathcal{F}_i, \quad \mathrm{Corr}\big(R(x), Q(x)\big) < \tau,$$

*where $\tau > 0$ is a correlation threshold and $\mathrm{Corr}(\cdot, \cdot)$ denotes the Pearson correlation coefficient.*

**Assumption 1** (Weak Reward Models Are Prone to Reward Hacking). *We hypothesize that weak reward models $R^W$ are prone to reward hacking, while strong reward models $R^S$ stabilize training.*

Weak reward models $R^W$ poorly correlate with human perceptual judgment, creating exploitable shortcuts. Rather than improving genuine perceptual quality which is a complex and multifaceted challenge, text-to-image models would maximize scores by targeting superficial features that these weak models overvalue. This leads to reward hacking, models chase artificial score improvements instead of authentic quality gains, ultimately degrading the generated images.

In contrast, strong reward models $R^S$ maintain consistent alignment with human preference, providing reliable guidance for stable training and ensuring the example towards meaningful quality improvements.

## 3 PARETO-FRONTIER-GUIDED OPTIMAL TRANSPORT

### 3.1 MULTI-REWARD OPTIMIZATION FORMULATION

Multi-reward optimization aims to achieve synergistic improvements across multiple reward models. Since the objectives of different rewards may inherently conflict, the problem naturally falls within the framework of Pareto optimization. In the Pareto optimization framework, "conflict" does not imply that different objectives are negatively correlated across the entire space; rather, it specifically refers to the inevitable trade-offs that arise on the Pareto frontier. In suboptimal regions of the solution space, multiple rewards may still improve simultaneously; however, as optimization approaches the frontier, further improving one reward necessarily requires sacrificing at least one other reward dimension. For example, image-text alignment reward and human preference reward may exhibit mutually reinforcing behavior in the early stage of optimization, but once either reward approaches its optimal regime, further improvement typically leads to degradation in other reward dimensions, which reflects the essence of Pareto-style conflict. We provide empirical evidence of such conflicts in Appendix C. We next introduce the fundamental concepts related to Pareto optimality under multiple reward signals.

**Pareto Optimality Fundamentals.** Given $K$ rewards $\tilde{R} = (R^1(x), R^2(x), \dots, R^K(x))$ to be maximized, the concepts of Pareto optimality can be defined as follows:

- **Pareto Dominance.** The reward vector of $x^a$ *dominates* that of $x^b$ (denoted $\tilde{R}(x^a) \succ \tilde{R}(x^b)$) if

$$\forall u \in \{1, \dots, K\} : R^u(x^a) \geq R^u(x^b), \quad \text{and} \quad \exists v \in \{1, \dots, K\} : R^v(x^a) > R^v(x^b).$$

- **Pareto Optimality.** For a given prompt $p_i$, a sample $x^* \in \mathcal{F}_i$ is *Pareto optimal* if its reward vector $\tilde{R}(x^*)$ is not dominated by that of any other $x \in \mathcal{F}_i$:

$$x^* \text{ is Pareto optimal } \iff x^* \in \mathcal{F}_i \text{ and } \nexists x \in \mathcal{F}_i : \tilde{R}(x) \succ \tilde{R}(x^*).$$

- **Pareto Front.** For a given prompt $p_i$, the *Pareto front* $\mathcal{J}_i$ is the set of all samples in the perceptually admissible feasible set $\mathcal{F}_i$ whose reward vectors are Pareto optimal :

$$\mathcal{J}_i = \{x \mid x \in \mathcal{F}_i, \nexists x' \in \mathcal{F}_i : \tilde{R}(x') \succ \tilde{R}(x)\}.$$

To tackle the issues beyond reward hacking identified in Section 2, we propose a Pareto-guided optimal transport framework for multi-reward optimization. First, an offline strategy leverages precomputed Pareto fronts for each prompt domain to mitigate reward hacking arising from heterogeneous bounds (Section 2.1). Second, addressing the assumption that weak rewards are prone to collapse (Assumption 1), we introduce a GPT-4o-based decision agent to detect and eliminate unstable weak rewards. Third, an online strategy enables strong rewards (Definition 2) to autonomously explore and expand the Pareto front during optimization. Finally, we propose JDR and JDS as metrics to evaluate performance gains and stability in multi-reward optimization.

### 3.2 OFFLINE PARETO FRONTIER GUIDANCE STRATEGY

To address the reward hacking issue caused by the global bound, as discussed in Proposition 1 of Section 2.1, we precompute prompt-specific candidate reward sets to estimate heterogeneous reward bounds before training. Specifically, for a given prompt $p_i$, the T2I model generates $M$ candidate

samples $\{x_i^j\}_{j=1}^M$, forming the precomputed candidate set $\mathcal{R}_{i,M}^{(\text{pre})} = \{\tilde{R}(x_i^j) \mid j = 1, \ldots, M\}$. We then adopt dominance matrix–based non-dominated sorting to extract the Pareto frontier. For the $M$ candidate reward vectors in $\mathcal{R}_{i,M}^{(\text{pre})}$, we build an $M \times M$ dominance matrix $A$, with $A_{mn} = 1$ if $x_i^m$ dominates $x_i^n$ (i.e., $\tilde{R}(x_i^m) \succ \tilde{R}(x_i^n)$), and 0 otherwise. Reward vectors with a domination count of zero (that is, those not dominated by any other vectors) constitute the Pareto frontier:

$$\mathcal{R}^{front}(p_i) = \{\tilde{R}(x_i^j) \in \mathcal{R}_{i,M}^{(\text{pre})} \mid \sum_{m=1}^{M} A_{mj} = 0\}, \tag{5}$$

where $q_i = |\mathcal{R}^{front}(p_i)|$ denotes the number of Pareto-optimal points on the Pareto frontier for prompt $p_i$. During training, for each prompt $p_i$, we retrieve its corresponding Pareto frontier $\mathcal{R}^{front}(p_i)$ and generate samples with the T2I model to obtain reward vectors dominated by all points on the frontier. The Pareto frontier serves as the optimization target, guiding these dominated rewards toward Pareto-optimal solutions. This is formalized by minimizing the optimal transport discrepancy between the distributions of dominated reward vectors and Pareto frontier reward vectors.

**Optimal Transport Framework.** Optimal Transport (OT) Monge (1781) provides a principled way to measure discrepancies between probability distributions while preserving the geometry of the underlying space. Given a source distribution $\mu$ and a target distribution $\nu$, OT seeks a transport plan $\gamma \in \Pi(\mu, \nu)$ that minimizes the total cost:

$$\min_{\gamma \in \Pi(\mu,\nu)} \int c(x, y) \, d\gamma(x, y), \tag{6}$$

where $c(x, y)$ denotes the ground cost between source sample $x$ and target sample $y$. Here, the $n$ rewards dominated by the frontier collectively constitute the source distribution: $\mu_i = \{\tilde{R}(x_i^j) \mid x_i^j \in \{x_i^1, \ldots, x_i^n\}, \, \forall \tilde{R}(x_i^m) \in \mathcal{R}^{(q)}(p_i) : \tilde{R}(x_i^m) \succ \tilde{R}(x_i^j)\}$, while the precomputed Pareto frontier serves as the target distribution $\nu_i = \mathcal{R}^{(q)}(p_i)$.

OT establishes a minimal-cost mapping from $\mu_i$ to $\nu_i$, transporting dominated samples toward dominating points according to geometric distances in the reward space for prompt $p_i$. Practically, the optimal plan $\gamma_i^*$ is solved by optimizing the following discrete formulation of OT with an entropy regularization term using the Sinkhorn algorithm Cuturi (2013):

$$\gamma_i^* = \arg \min_{\gamma \in \Pi(\mu_i, \nu_i)} \sum_{m,j} c(y_i^m, x_i^j) \, \gamma(y_i^m, x_i^j), \tag{7}$$

where the ground cost is defined as $c(y_i^m, x_i^j) = \|\tilde{R}(y_i^m) - \tilde{R}(x_i^j)\|_2^2$, representing the squared Euclidean distance between reward vectors of source sample $y_i^m$ and target sample $x_i^j$ whose reward vector is in the Pareto frontier, and the $\gamma$ is a $n \times q_i$ transport matrix.

This offline strategy is applicable to multi-reward training scenarios containing suboptimal reward models prone to shortcut behaviors, providing the model with precomputed Pareto frontiers as optimization targets to guide collaborative robust optimization across multiple reward models.

### 3.3 GPT-4O BASED DECISION-MAKING AGENT

Our experiments confirm the hypothesis (Section 2.2, Assumption 1) that weak reward models are susceptible to reward hacking. To mitigate this, we employ **GPT-4o** as a decision-making agent for detecting and removing collapsed weak reward models. Once significant signals of reward hacking occur, the optimization trajectory becomes difficult to recover, making early detection critical. However, in the early stages of reward hacking, the generated images are only mildly collapsed, with only slight differences from normally generated images.

To capture these subtle differences, we not only collected images of early mild collapses for each reward model but also utilized GPT-4o to analyze their respective collapse characteristics, constructing a comprehensive prior reference for the agent. Upon detecting mild collapse, the agent immediately removes the problematic weak reward model and reverts to the earlier stable checkpoint to safely resume training. A detailed workflow and description of the agent are provided in Appendix D.

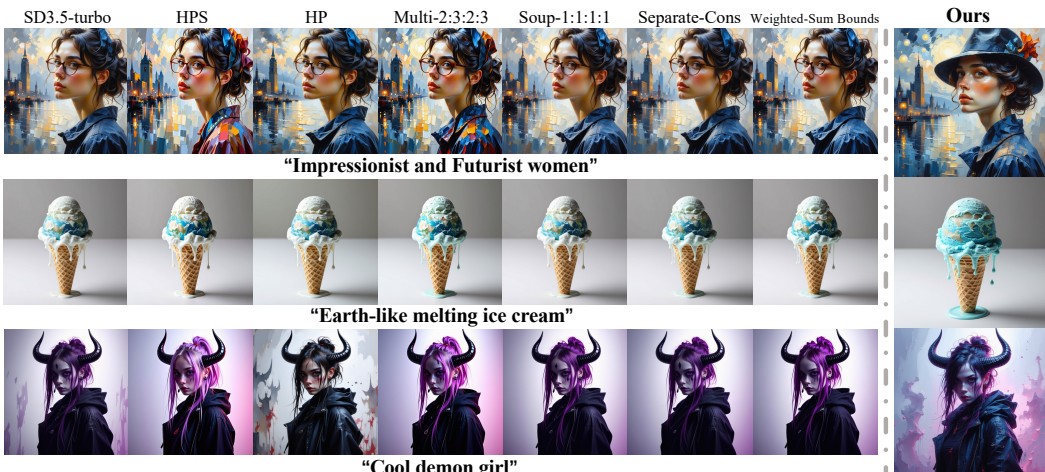

Figure 1: Qualitative Comparison of Optimization Results Across Different Methods.

### 3.4 ONLINE PARETO DOMINATION STRATEGY

After the decision-making agent eliminates all weak reward models (Definition 3) prone to reward hacking, the remaining strong reward models (Definition 2) exhibit a strong correlation between their preference prediction capabilities and human perceptual quality. To further explore superior reward bounds, we propose an online strategy that enables strong reward models to autonomously collect and optimize along the Pareto frontier during training. Specifically, given a prompt $p_i$, we perform the T2I model in parallel across multiple processes to increase the number of generated images. The reward vectors from all processes are aggregated into a global set $\mathcal{R}_{i,N} = \{\tilde{R}(x_i^j) \mid j = 1, \ldots, N\}$, where $n$ is the number of candidate images per process, $P$ is the number of processes, and $N = P \times n$.

For each sample $x_i^j$, we identify all vectors in $\mathcal{R}_{i,N}$ that Pareto-dominate $\tilde{R}(x_i^j)$:

$$\mathcal{R}^{dom}(x_i^j) = \{\tilde{R}(x_i^m) \in \mathcal{R}_{i,N} \mid \tilde{R}(x_i^m) \succ \tilde{R}(x_i^j)\}, \tag{8}$$

where $k_i = |\mathcal{R}^{dom}(x_i^j)|$ is the number of dominating reward vectors. The reward vector of $x_i^j$ forms a discrete source distribution with one sampling point $\mu = \tilde{R}(x_i^j)$, while the dominating set serves as target distribution $\nu = \mathcal{R}^{dom}(x_i^j)$. For each batch, the optimal transport discrepancies between the source and target distributions for all samples are computed and summed to obtain the batch loss.

As training progresses and the reward candidate set improves, the online strategy adaptively guides strong reward models to explore superior Pareto frontiers, thereby achieving better optimization.

### 3.5 JOINT PERFORMANCE METRICS

Using the mean of individual rewards as an evaluation metric has clear limitations: improvements in average reward values do not guarantee per-sample gains across all reward dimensions, and reward hacking may cause reward values to rise while human-perceived quality declines, thereby obscuring the true optimization dynamics. To address this, we propose the **Joint Domination Rate (JDR)**, which requires simultaneous improvement over the optimized baseline model across all $K$ rewards:

$$\text{JDR}_K = \frac{1}{N} \sum_{i=1}^{N} \mathbb{1}\!\!\!\!\,/\left( \tilde{R}_i^j > \tilde{R}_i^{j,(b)}, \; \forall j \in \{1, \ldots, K\} \right), \tag{9}$$

where $\mathbb{1}\!\!\!\!\,/(\cdot)$ denotes the indicator function, and $R_i^j$ and $R_i^{j,(b)}$ are the current and baseline scores of sample $i$ on reward $R^j$. Similarly, we define the **Joint Collapse Rate (JCR)** to measure the proportion of samples that degrade across all rewards $\text{JCR}_K = \frac{1}{N} \sum_{i=1}^{N} \mathbb{1}\!\!\!\!\,/\left( \tilde{R}_i^j < \tilde{R}_i^{j,(b)}, \; \forall j \in \{1, \ldots, K\} \right)$. By enforcing all-dimensional consistency, JDR and JCR effectively avoid statistical illusions and provide more reliable evaluation for multi-reward optimization.

Table 1: Quantitative Results (%) on the Parti-Prompts dataset Yu et al. (2022): Individual Reward Win Rates and Joint Performance Metrics Relative to SD3.5-Turbo.

| Model | ICT (%)↑ | HP (%)↑ | CLIP (%)↑ | HPS (%)↑ | JDR$_2$ (%)↑ | JDR$_4$ (%)↑ | JCR$_4$ (%)↓ |
|---|---|---|---|---|---|---|---|
| **Single-Reward Optimization** | | | | | | | |
| + ICT | 56.99 | 36.83 | 47.06 | 48.71 | 20.59 | 7.66 | 10.17 |
| + HP | 52.45 | **90.26** | 44.30 | 57.29 | _36.15_ | _13.73_ | 4.11 |
| + CLIP | 48.96 | 47.06 | **52.63** | 46.81 | 23.77 | 8.09 | 9.07 |
| + HPS | 50.12 | 41.67 | 37.07 | **88.30** | 20.77 | 8.03 | 3.06 |
| **Multi-Reward Joint Optimization** (Weighted **ICT:HP:CLIP:HPS** ratios) | | | | | | | |
| 1:1:1:1 | 51.10 | 52.08 | _47.43_ | 82.97 | 26.59 | 12.68 | 3.19 |
| 2:3:2:3 | 50.80 | 56.43 | 46.51 | _86.03_ | 28.31 | 13.42 | _2.57_ |
| 2:2:3:3 | 50.43 | 56.56 | 46.57 | 84.25 | 26.23 | 12.62 | 2.76 |
| 4:4:1:1 | 51.96 | 57.23 | 44.12 | 79.90 | 29.53 | 12.44 | 3.74 |
| **Reward Soup** (Weighted **ICT:HP:CLIP:HPS** fusion of single-reward LoRAs) | | | | | | | |
| 1:1:1:1 | 50.55 | 54.17 | 42.16 | 81.92 | 27.02 | 11.15 | 3.74 |
| 1:1:4:4 | 50.43 | 52.94 | 42.46 | 85.11 | 25.37 | 11.15 | 3.31 |
| 3:2:1:4 | 50.80 | 53.74 | 43.32 | 85.29 | 26.29 | 10.85 | 3.19 |
| 3:2:0:0 | 50.74 | 53.86 | 42.59 | 83.21 | 26.10 | 10.85 | 3.31 |
| **Multi-Reward under Heterogeneous Bounds** | | | | | | | |
| Weighted-Sum | 52.63 | 56.86 | 46.94 | 82.48 | 29.84 | 13.66 | 3.49 |
| Separate-Cons | 49.45 | 57.23 | 46.63 | 61.21 | 28.25 | 10.78 | 6.68 |
| **Pareto-Frontier-Guided Optimal Transport** | | | | | | | |
| **Ours** | _56.43_ | _85.23_ | 43.63 | 61.70 | **47.98** | **17.10** | **2.39** |

## 4 EXPERIMENTS

### 4.1 EXPERIMENT SETTING

**T2I Model.** Our text-to-image (T2I) framework builds upon Stable Diffusion 3.5-Turbo, one of the most advanced text-to-image models. To maintain the stability of the base model, we update only LoRA Hu et al. (2021) parameters rather than the original weights of Stable Diffusion 3.5-Turbo. Specifically, we employ the gradient backpropagation strategy of DRaFT-K Clark et al. (2024) for fine-tuning, which applies gradient updates only during the final $k = 2$ denoising steps to reduce memory consumption and accelerate training. Training details are in Appendix G.

### 4.2 BASELINE CONSTRUCTION

To fairly evaluate the effectiveness of the proposed multi-reward optimization framework, we construct four representative baseline methods: single-reward fine-tuning, weighted multi-reward fine-tuning, reward soup, and multi-reward fine-tuning under heterogeneous reward bounds. We provide a detailed description of the baselines related to heterogeneous upper bounds here, and present the standard baseline configurations in Appendix G.

**Multi-Reward Fine-Tuning with Heterogeneous Reward Bounds.**

We utilize the precomputed Pareto frontier as an approximation of prompt-wise heterogeneous reward bounds and construct three baseline variants upon it. The first variant aggregates approximate bounds of multiple reward functions into a weighted average, serving as a unified optimization target across prompts, denoted as the *weighted-sum bounds* method. The second variant assigns each reward bound as its optimization target and minimizes the squared error between each reward and its bound, formulated as $\mathcal{L} = \sum_k (r_k^{\text{bound}} - r_k)^2$, denoted as the *separate-constraints* method. The third variant leverages the online-computed Pareto frontier as the optimization targets for each reward and employs a simple mapping loss, referred to as the *Frontier-Mapping* method, which we present in Appendix E.

### 4.3 EVALUATION AND ANALYSIS

**Joint Performance Metrics.** In Table 1, we present the individual win rates of the four reward models (ICT, HP, CLIP, and HPS) together with the joint optimization performance on the Parti-Prompts dataset Yu et al. (2022). Our method consistently demonstrates significant improvements over the baselines, achieving an 11% gain in JDR$_2$, a 3.4% gain in JDR$_4$, and a 0.2% reduction in JCR$_4$, while maintaining comparable win rates on each individual reward. Single-reward baselines achieve

Table 2: Quantitative Results on the Parti-Prompts dataset: Comparison of multiple reward scores.

| Model | CLIP↑ | ICT↑ | Aesthetic↑ | HPS↑ | PickScore↑ | ImgReward↑ | HP↑ |
|---|---|---|---|---|---|---|---|
| SD3.5-Turbo | 0.3372 | 0.8965 | 6.2766 | 0.2856 | 22.7435 | 1.1499 | 0.7754 |
| **Single-Reward Optimization** | | | | | | | |
| ICT | 0.3548 | 0.8986 | 6.2781 | 0.2916 | 22.7261 | 1.1487 | 0.7739 |
| HP | 0.3538 | 0.8976 | 6.2881 | 0.2809 | 22.7687 | 1.1691 | 0.7776 |
| CLIP | **0.3554** | 0.8958 | 6.2770 | 0.2915 | 22.7396 | 1.1530 | 0.7751 |
| HPS | 0.3501 | 0.8987 | 6.3437 | **0.2994** | 22.6896 | 1.1793 | 0.7747 |
| **Multi-Reward Joint Optimization** (Weighted **ICT:HP:CLIP:HPS** ratios) | | | | | | | |
| 1:1:1:1 | 0.3545 | 0.8970 | 6.2839 | 0.2958 | 22.6895 | 1.1464 | 0.7763 |
| 2:3:2:3 | 0.3544 | 0.8987 | 6.2850 | 0.2971 | 22.6835 | 1.1636 | 0.7763 |
| 2:2:3:3 | 0.3552 | 0.8974 | 6.2787 | 0.2962 | 22.6926 | 1.1570 | 0.7770 |
| 4:4:1:1 | 0.3541 | 0.8989 | 6.3043 | 0.2950 | 22.7049 | 1.1568 | 0.7762 |
| **Reward Soup** (Weighted **ICT:HP:CLIP:HPS** fusion of single-reward LoRAs) | | | | | | | |
| 1:1:1:1 | 0.3543 | 0.8958 | 6.2931 | 0.2936 | 22.7542 | 1.1562 | 0.7752 |
| 1:1:4:4 | 0.3539 | 0.8967 | 6.3037 | 0.2951 | 22.7559 | 1.1679 | 0.7752 |
| 3:2:1:4 | 0.3537 | 0.8965 | 6.3032 | 0.2951 | 22.7549 | 1.1654 | 0.7754 |
| 3:2:0:0 | 0.3541 | 0.8961 | 6.2759 | 0.2941 | 22.7436 | 1.1493 | 0.7752 |
| **Multi-Reward under Heterogeneous Bounds** | | | | | | | |
| Weighted-Sum | 0.3546 | 0.8968 | 6.2800 | 0.2947 | 22.7274 | 1.1514 | 0.7760 |
| Separate-Cons | 0.3549 | 0.8967 | 6.2774 | 0.2922 | 22.7343 | 1.1561 | 0.7757 |
| **Pareto-Frontier-Guided Optimal Transport** | | | | | | | |
| **Ours** | 0.3534 | **0.9004** | **6.3588** | 0.2929 | **22.8160** | **1.1808** | **0.7783** |

Figure 2: Pareto Frontier Visualization based on strong rewards (ICT and HP) on Three Prompts.

the highest win rates on their respective rewards, but their joint performance metrics generally degrade. Multi-reward optimization baselines that rely on the global bound perform worse than single-reward optimization with HP, consistent with Assumption 1 in Section 2.2, which states that weak rewards in multi-reward optimization induce collapse and undermine joint performance.

**Reward Metrics.** Table 2 presents comprehensive evaluation results across seven reward models. Our method consistently secures the best performance on most metrics, demonstrating robustness under heterogeneous bounds and strong-reward training. In contrast, multi-reward joint optimization methods are affected by weak rewards, leading to degraded overall performance and instability.

**Quantitative results.** As shown in Fig. 1, we present the qualitative comparisons. The baselines include two single-reward optimizations aligned with human preference, HPS and HP; a multi-reward joint training approach with the global bound as the optimization target using the weighted ratio ICT:HP:CLIP:HPS = 2:3:2:3; a reward-soup fusion method applied at inference with equal weighting ICT:HP:CLIP:HPS = 1:1:1:1; and two heterogeneous-bound methods, Weighted-Sum bounds and Separate-Constraints. The results demonstrate that our method achieves superior visual quality while avoiding reward hacking. Additional qualitative examples are provided in Appendix F.

**Qualitative Case Studies on Pareto Frontier Visualization.** We conduct a Pareto frontier analysis based on two strong rewards, ICT and HP, comparing SD 3.5-Turbo, the HP-optimized single-reward model, the multi-reward joint optimization model with ratio ICT:HP:CLIP:HPS = 2:3:2:3, and our proposed method. For each prompt and identical random seed, every method generates 50 images, whose reward distributions and corresponding Pareto frontiers are plotted in a two-dimensional diagram. As shown in Fig. 2, the image domains induced by different prompts exhibit heterogeneous reward bounds. Our method consistently produces Pareto frontiers that dominate those of the

Table 3: Win Rate (%) of Ours vs. Baselines by Human Experts on DiffusionDB Wang et al. (2023).

| Model | SD3.5-Turbo | HP | HPS | Soup-1:1:1:1 | Multi-2323 | Separate-Constraints | Weighted-Sum Bounds |
|---|---|---|---|---|---|---|---|
| **Ours** vs. | 76.34 | 61.29 | 74.19 | 74.19 | **79.57** | 68.82 | 73.12 |

Figure 3: Comparative Training Curves of Joint Domination Rate ($JDR_4$) for Ours versus Baseline Methods.

| Model | $JDR_2\uparrow$ | $JDR_4\uparrow$ | $JCR_4\downarrow$ |
|---|---|---|---|
| SD3.5-Turbo | – | – | – |
| Online Only | 21.51 | 9.38 | 4.04 |
| Offline Only | 34.07 | 7.35 | 7.23 |
| Ours (w/o Online) | 38.54 | 14.89 | 4.29 |
| **Ours** | **47.98** | **17.10** | **2.39** |

Table 4: Ablation Study on Joint Domination Rate (JDR) and Joint Collapse Rate (JCR).

baselines, with generated samples distributed closer to the frontier, reflecting superior trade-off alignment and validating the effectiveness of our approach. More visualizations in Appendix F.

**Visualization and Analysis of Training Curve.** In Figure 3, we present the training curves of the Joint Domination Rate ($JDR_4$), which serves as a robust metric of joint optimization, for our method and the baselines. Single-reward baselines optimized with the global bound collapse rapidly, while multi-reward optimization with the global bound and the involvement of weak rewards also exhibits severe deterioration. In contrast, our method achieves stable and consistent improvements throughout training while effectively avoiding reward hacking.

Table 5: Agreement accuracy between VLM-based reward hacking detection and human expert assessments across 200 evaluation samples.

| Model | GPT-4o | Qwen3-VL-32B-Thinking | GLM-4.5V | Qwen3-VL-8B-Thinking |
|---|---|---|---|---|
| Accuracy | 90.5% | 87.5% | 84.0% | 83.5% |

**Reproducibility, Accessibility, and Cost Analysis of the Agent.** As shown in Table 5, GPT-4o achieves a 90.5% agreement with human experts on 200 evaluation samples, demonstrating strong reproducibility. Leveraging the model-agnostic design of our framework, we further employ several powerful open-source VLMs, including Qwen3-VL-32B-Thinking Bai et al. (2025), GLM-4.5V Team et al. (2025), and Qwen3-VL-8B-Thinking Bai et al. (2025), as alternative agents. These models achieve up to 87% agreement with human experts, confirming that accessible open-source models can reliably replace GPT-4o without compromising performance. Finally, since the agent is invoked only at fixed intervals during training, the computational cost remains minimal. Specifically, each detection pass costs about $0.015, and the overall overhead is only 0.4% of the total training cost.

**User study.** We conduct a user study with ten annotators on 300 randomly selected prompts from the DiffusionDB dataset Wang et al. (2023). For each prompt, image pairs (ours vs. baseline) were presented in random order, and annotators evaluated prompt fidelity and visual appeal. As shown in Table 3, our method achieves higher win rates against all baselines, confirming its effectiveness.

**Ablation Study** We conduct ablation experiments on the online policy, offline policy, and GPT-4o Agent, as shown in Table 4. It can be observed that using the online or offline policy alone yields moderate gains. Excluding the online policy while combining the offline policy with the GPT-4o Agent achieves the second-best results, where strong reward models mitigate collapse and approach the precomputed bound. Finally, our approach, incorporating the offline policy, adaptive regulation by the GPT-4o Agent, and exploration through the online policy, achieves optimal performance. Furthermore, to validate the robustness of our hyperparameter selection, we provide ablation experiments on the entropy regularization strength $\varepsilon$ in the Appendix E.

## 5 CONCLUSION

In this work, we demonstrate that reward hacking arises from unified global targets under heterogeneous reward bounds, and from the inherent vulnerability of weak reward models. To address this, we propose a Pareto-frontier-guided optimal transport framework with online and offline strategies, and introduce JDR and JCR as principled evaluation metrics. Our approach achieves consistent gains over strong baselines while effectively mitigating reward hacking.

REPRODUCIBILITY STATEMENT

The use of LLMs is detailed in App. H. All models and training setups, along with their complete hyperparameter configurations, are described in Sec. 4.1 and App. G. To comply with institutional policies and preserve double-blind review, implementation resources will be released only after the review period. Upon publication, we will release the full codebase under an open license to ensure full reproducibility.

ETHICS STATEMENT

This work adheres to the ICLR Code of Ethics. All datasets used in this study are publicly available or synthetically generated and do not contain personally identifiable information. The proposed method is intended solely for research purposes, aiming to enhance the capability of generative models in handling multi-reward objectives and does not intend to promote harmful or biased content. We will release our code and data under a research-friendly license to ensure transparency, reproducibility, and responsible use.

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

## A RELATED WORK

**Single-Reward Optimization for Text-to-Image Diffusion Models**

Reward-based optimization has emerged as a crucial paradigm for improving text-to-image diffusion models, where reward models provide supervision signals to enhance generation quality. Current research focuses on two objectives: **(1) Text–Image Alignment.** CLIP Radford et al. (2021) pioneered cross-modal embeddings that capture vision–language semantics, and BLIP Li et al. (2022) extended this paradigm with bidirectional mechanisms for stronger alignment evaluation. However, ICT Ba et al. (2025) demonstrated that both models systematically undervalue high-quality images, motivating the development of refined metrics for more faithful text representation. **(2) Human Preference Alignment.** Recent efforts have shifted toward modeling human perceptual preferences as rewards to better align generation with subjective judgments. Reward models such as ImageReward Xu et al. (2023), HPS Wu et al. (2023), PickScore Kirstain et al. (2023), and HP Ba et al. (2025) are trained on large-scale preference datasets to approximate human judgments. However, these models may yield conflicting signals. A central challenge, therefore, lies in how to effectively integrate and jointly optimize multiple rewards to reconcile such discrepancies.

**Multi-Reward Optimization and Pareto-Front Methods for Text-to-Image Diffusion**

To simultaneously improve text–image consistency and human preference alignment, previous studies have introduced multiple reward signals into text-to-image diffusion models. Basic approaches typically rely on linear weighting; for example, ReNO Eyring et al. (2024) applies weighted fusion of multiple rewards solely to optimize the noise initialization stage, while TextCraftor Li et al. (2024a) confines the weighted optimization to the text encoder. However, such simple weighting schemes are limited when handling conflicting rewards. To address this, several works have introduced the concept of the Pareto-Optimal: Parrot Lee et al. (2024) refines prompts via a Prompt Expansion Network and selects sample pairs that dominate across all rewards for reinforcement learning; within diffusion-DPO Wallace et al. (2024) frameworks, CaPO Lee et al. (2025) calibrates different rewards to enhance training stability, and BalancedPO Tamboli et al. (2025) aggregates diverse reward signals through majority voting. Nevertheless, these methods often require additional auxiliary modules or extensive relabeling of paired samples, and they have not thoroughly examined the underlying causes of reward hacking—the most critical issue in reward optimization. In contrast, our method eliminates the need for paired datasets, directly integrates multiple rewards, and employs optimal transport to guide batch samples toward the Pareto frontier, thereby achieving more balanced and robust alignment under conflicting objectives.

## B PROOFS THAT GLOBAL UPPER BOUNDS INDUCE REWARD HACKING

**Standing assumptions.** We introduce two weak regularity conditions to formalize the role of the feasible set $\mathcal{F}_i$ associated with each prompt $p_i$:

**(A1) Tightness on $\mathcal{F}_i$.** For each image domain $\mathcal{D}_i$ associated with prompt $p_i$, the reward upper bound $\overline{R_i} := \sup_{x \in \mathcal{F}_i} R(x)$ (or, in the multi-reward case, $\overline{R_i^k} := \sup_{x \in \mathcal{F}_i} R^k(x)$ for each $k$) is tight for the admissible set $\mathcal{F}_i$.

**(A2) Admissibility gap.** The admissible set $\mathcal{F}_i$ consists of perceptually acceptable samples, where $\underline{Q_i} := \inf_{x \in \mathcal{F}_i} Q(x)$ exists. Samples outside $\mathcal{F}_i$ may either be perceptually comparable or inferior; in particular, those falling into the reward hacking region are characterized by spuriously high reward values yet degraded perceptual quality, i.e., $Q(x) < \underline{Q_i}$.

### B.1 SINGLE-REWARD CASE

*Proof of Proposition 1 (single reward) by contradiction.* Let a prompt $p_i$ induce image domain $\mathcal{D}_i$ and admissible set $\mathcal{F}_i \subseteq \mathcal{D}_i$. Let $C = \sup_{i, x \in \mathcal{F}_i} R(x)$ be the global constant used in the surrogate loss $\mathcal{L}(x) = C - R(x)$. By Property 1 and global construction of $C$, we have $C > \overline{R_i}$ for some $p_i$.

**Assume for contradiction** that there exists $x \in \mathcal{X}$ such that

$$\mathcal{L}(x) < C - \overline{R_i} \quad \text{and} \quad x \notin \mathcal{H}_i.$$

The loss inequality is equivalent to $R_i(x) > \overline{R_i}$. By **(A1)**, any point with $R_i(x) > \overline{R_i}$ cannot lie in $\mathcal{F}_i$, hence $x \notin \mathcal{F}_i$.

Now consider two possibilities:

1. If $x \notin \mathcal{D}_i$, then $x$ is not a sample produced for prompt $p_i$, contradicting that we analyze optimization within $\mathcal{D}_i$.

2. If $x \in \mathcal{D}_i$, then from $R(x) > \overline{R_i}$ and the definition of $\overline{R_i}$ we have $x \notin \mathcal{F}_i$. By Definition 1, any sample outside $\mathcal{F}_i$ with $R(x) > \overline{R_i}$ must also satisfy $Q(x) < \underline{Q_i}$. Therefore, $x$ exhibits reward hacking, i.e., $x \in \mathcal{H}_i$.

Both cases contradict the assumption. Hence, any step achieving $\mathcal{L}(x) < C - R_i^\star$ necessarily produces $x \in \mathcal{H}_i$. This proves the claim for the single-reward case. $\qquad\square$

### B.2 MULTI-REWARD CASE

*Proof of Proposition 1 (multi reward) by contradiction.* Fix a prompt $p_i$ with domain $\mathcal{D}_i$ and admissible set $\mathcal{F}_i$. Let rewards be $\tilde{R}(x) = (R^1(x), \dots, R^K(x))$, with weights $w_k \geq 0$ satisfying $\sum_{k=1}^{K} w_k = 1$. Define the admissible weighted bound

$$\overline{S_i} := \sup_{x \in \mathcal{F}_i} \sum_{k=1}^{K} w_k R^k(x), \qquad \overline{R_i^k} := \sup_{x \in \mathcal{F}_i} R^k(x).$$

Let the global constant $C$ in $\mathcal{L}(x) = C - \sum_{k=1}^{K} w_k R^k(x)$ be constructed from cross-domain upper bounds (Sec. 2.1), so that $C > \overline{S_i}$ for some $i$.

**Assume for contradiction** that there exists $x \in \mathcal{X}$ such that

$$\mathcal{L}(x) < C - \overline{S_i} \quad \text{and} \quad x \notin \mathcal{H}_i.$$

The loss inequality is equivalent to

$$\sum_{k=1}^{K} w_k R^k(x) > \overline{S_i}.$$

By definition of $\overline{S_i}$, no point in $\mathcal{F}_i$ can exceed this value, hence $x \notin \mathcal{F}_i$. Moreover, since the weighted sum strictly exceeds $\overline{S_i}$, there must exist at least one coordinate $k'$ such that

$$R^{k'}(x) > \overline{R_i^{k'}}.$$

Now consider two possibilities:

1. If $x \notin \mathcal{D}_i$, then $x$ is not a sample produced for prompt $p_i$, contradicting that we analyze optimization within $\mathcal{D}_i$.

2. If $x \in \mathcal{D}_i$, then from $R^{k'}(x) > \overline{R_i^{k'}}$ and the definition of $\overline{R_i^{k'}}$ we have $x \notin \mathcal{F}_i$. By Definition 1, any sample outside $\mathcal{F}_i$ with some reward dimension exceeding its admissible upper bound, i.e. $R^{k'}(x) > \overline{R_i^{k'}}$, must also satisfy $Q(x) < \underline{Q_i}$. Therefore, $x$ exhibits reward hacking, i.e., $x \in \mathcal{H}_i$.

Both cases contradict the assumption. Hence, any step achieving $\mathcal{L}(x) < C - \overline{S_i}$ necessarily produces $x \in \mathcal{H}_i$. This proves the claim for the multi-reward case. $\qquad\square$

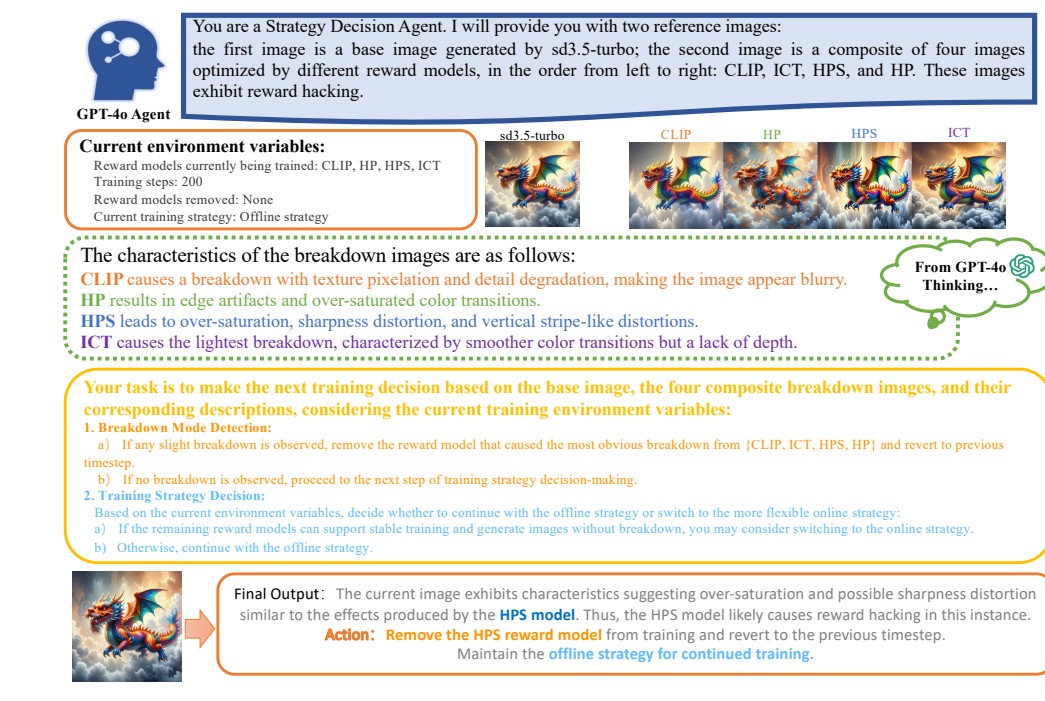

Figure 4: Adaptive Decision Pipeline of the GPT-4o based Agent for Multi-Reward Optimization.

## C    EXPERIMENTAL EVIDENCE OF REWARD CONFLICTS

To demonstrate the conflict between image-text alignment reward and human preference reward, we optimize Stable Diffusion 3.5-Turbo using the CLIP model and report the evolution of image-text alignment score and human preference score throughout training in Table 6. When optimizing solely for CLIP, the image-text alignment scores (CLIP/ICT) improve by +7.3% / +6.2%, while the human preference scores (HPS/HP) degrade by –2.8% / –4.4%. This clearly illustrates the inherent conflicts between these reward objectives.

Table 6: Image-text alignment and human preference score trends during CLIP-only optimization.

| Metric | Reward Type | Base | 100 | 200 | 300 | 400 | Trend |
|---|---|---|---|---|---|---|---|
| CLIP | Image-Text Alignment | 0.2943 | 0.2944 | 0.3043 | 0.3136 | 0.3157 | ↑ |
| ICT | Image-Text Alignment | 0.7738 | 0.7738 | 0.7758 | 0.8082 | 0.8215 | ↑ |
| HPS | Human Preference | 0.2558 | 0.2558 | 0.2556 | 0.2492 | 0.2487 | ↓ |
| HP | Human Preference | 0.7789 | 0.7789 | 0.7785 | 0.7711 | 0.7448 | ↓ |

## D    DETAILS OF GPT-4O BASED DECISION-MAKING AGENT

We introduce **GPT-4o** as a decision-making agent to adaptively manage multi-reward model training. The agent dynamically determines actions based on generated image quality and training stability, with three core capabilities:

- **Continue Training** — When no signs of collapse are observed in the generated images, the agent performs no additional operations and allows training to continue.
- **Remove & Revert** — Upon detecting a breakdown, the agent removes the unstable reward model and reverts to the most recent stable checkpoint.
- **Switch Strategy** — When training has proceeded for an extended period with minimal improvement in image optimization, while the remaining reward models remain stable, the agent smoothly transitions the process from offline training to online training.

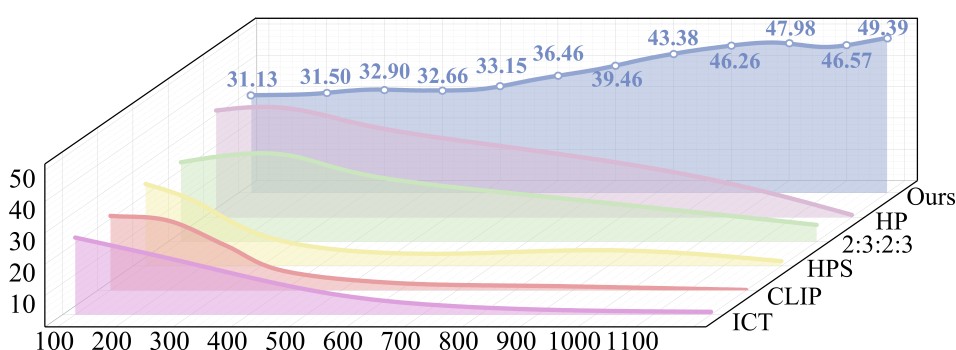

Figure 5: Training Curves of Joint Domination Rate (JDR$_2$).

## D.1 PRIOR KNOWLEDGE AND INITIALIZATION

**Knowledge Base Construction.** For each reward model, we collect degraded image samples generated under breakdown conditions and employ **GPT-4o** for semantic analysis to extract characteristic failure patterns. These include:

- **CLIP** — Texture pixelation and detail degradation, resulting in blurred appearance.
- **ICT** — Mildest breakdown, characterized by overly smooth transitions and loss of depth.
- **HPS** — Severe over-saturation, sharpness distortion, and vertical stripe artifacts.
- **HP** — Pronounced edge artifacts and over-saturated color transitions.

The extracted patterns are structured into a prior knowledge base, which supports subsequent chain-of-thought (CoT)-based decision-making.

**State Initialization.** At the beginning of training, the agent is initialized with the following state information: (i) current training strategy (offline/online), (ii) active reward model set $\{R^i\}$, (iii) cumulative training step count $n$, and (iv) historical stability metrics with corresponding checkpoint references.

## D.2 DECISION-MAKING WORKFLOW

**Environment Context.** At each decision step, the agent receives structured contextual information, including: (i) the baseline image generated by the underlying model; (ii) composite breakdown images corresponding to different reward models (CLIP, ICT, HPS, HP); (iii) environment variables such as the set of active reward models, training step count, removal history, and the current strategy state (offline/online).

**Decision Procedure.** The agent performs a two-stage reasoning process:

1. **Breakdown Detection.** Compare diagnostic images with prior templates.
   - If a breakdown is detected, remove the most severely affected reward model and revert to the previous checkpoint.
   - If no breakdown is detected, proceed to the strategy evaluation stage.
2. **Training Strategy Decision.** Based on the current environment variables:
   - If the remaining reward models remain stable without breakdown, switch from the offline strategy to the more flexible online strategy.
   - Otherwise, maintain the offline training strategy.

**Final Output.** The decision agent produces a standardized output, which includes:

```
Final Output: [Training state analysis]
Action: [Remove / Revert / Continue]
Strategy Maintenance: [Offline / Online]
```

## D.3   Detailed Training Procedure

We implemented a staged dynamic optimization approach with systematic evaluation checkpoints every 100 steps to ensure stability in multi-reward training. The procedure consisted of two main phases: an initial offline phase with joint reward model training, during which weaker reward models were progressively removed until only stable ones remained, followed by the activation of the online strategy once convergence was achieved.

**Offline Training with Adaptive Reward Management**   In the offline phase, four reward models (CLIP, ICT, HPS, HP) were trained simultaneously. At each 100-step interval, the GPT-4o decision agent performed automated diagnostic analysis on composite output images to detect potential training instabilities. When anomalies were detected, the system reverted to the most recent stable checkpoint and adjusted the active reward set before resuming training.

The first breakdown occurred at step 200, where evaluation revealed over-saturation artifacts, sharpness distortions, and vertical stripe patterns consistent with HPS model instability. The agent removed HPS from the active reward set and rolled training back to step 100, after which training continued with the reduced set (CLIP, ICT, HP).

Training proceeded stably through steps 300, 400, and 500 without anomalies. However, at step 600, diagnostic analysis identified characteristic texture pixelation and detail degradation in CLIP outputs, while ICT and HP remained stable. The agent isolated CLIP as the instability source, removed it from the active set, and reverted to the step 500 checkpoint. Training then continued with the remaining reward pair (ICT, HP).

**Transition to Online Training**   Between steps 500 and 800, training exhibited sustained stability with no further breakdowns. After confirming at step 800 that generated images showed no collapse and only minimal changes, the system initiated the transition to online training mode for further exploration.

## D.4   Limitations of Heuristic Statistical Methods

**Evaluation of Heuristic Methods for Detecting Reward Hacking.** We conducted comprehensive experiments to evaluate whether heuristic methods could effectively detect reward hacking. Specifically, we analyzed three common statistical indicators: mean reward, reward variance (standard deviation), and KL divergence from a reference distribution. We evaluated these metrics under two scenarios: (1) images generated from different prompts at the same training step (Table 7), and (2) images generated from the same prompt across different training steps (Table 8).

Our results demonstrate that **no reliable heuristic pattern emerges** from these statistics:

**(1) Cross-prompt heterogeneity (Table 7):** We generated 50 images per prompt at step 300 (where reward hacking has occurred) and found that different prompts exhibit drastically different reward statistics. Since each training step samples different prompts, these statistical measures are fundamentally unstable and cannot serve as a basis for effective heuristic detection.

**(2) Reward indistinguishability (Table 8):** We measured reward statistics on images generated from the same prompt across different training steps (50 images per prompt), with human expert evaluation showing reward hacking caused by HPS beginning at step 200. The results reveal critical limitations of heuristic methods:

**Mean and standard deviation fail to detect anomalies (Tables 8(a) and (b)):** Neither metric exhibits clear abnormal patterns at the collapse point, making them unreliable for detecting.

**KL divergence detects collapse but cannot disentangle rewards (Table 8(c)):** While KL divergence shows sharp spikes that could signal reward hacking, all rewards exhibit similar sudden changes simultaneously. This makes it impossible to identify which reward causes the collapse.

In contrast, our knowledge-driven agent successfully addresses both challenges: detecting when reward hacking occurs and identifying which reward causes it. These comprehensive experiments confirm that simple heuristic methods based on reward statistics are insufficient for detecting reward hacking in text-to-image generation, thereby motivating and validating our agent-based approach.

Table 7: Reward statistics exhibit heterogeneity across different prompts at training step 300.

<table>
<tr><td colspan="4">(a) Mean reward.</td><td colspan="4">(b) Standard deviation.</td><td colspan="4">(c) KL divergence.</td></tr>
<tr><th>Reward</th><th>p1</th><th>p2</th><th>p3</th><th>Reward</th><th>p1</th><th>p2</th><th>p3</th><th>Reward</th><th>p1</th><th>p2</th><th>p3</th></tr>
<tr><td>ICT</td><td>0.8676</td><td>0.8114</td><td>0.9095</td><td>ICT</td><td>0.09</td><td>0.09</td><td>0.06</td><td>ICT</td><td>8.00</td><td>5.45</td><td>11.42</td></tr>
<tr><td>CLIP</td><td>0.2205</td><td>0.1421</td><td>0.2956</td><td>CLIP</td><td>0.03</td><td>0.04</td><td>0.03</td><td>CLIP</td><td>13.18</td><td>19.48</td><td>4.89</td></tr>
<tr><td>HPS</td><td>0.2449</td><td>0.2661</td><td>0.2749</td><td>HPS</td><td>0.007</td><td>0.01</td><td>0.007</td><td>HPS</td><td>13.09</td><td>10.05</td><td>14.34</td></tr>
<tr><td>HP</td><td>0.7800</td><td>0.7805</td><td>0.7805</td><td>HP</td><td>0.0006</td><td>0.0009</td><td>0.0005</td><td>HP</td><td>11.04</td><td>10.78</td><td>16.82</td></tr>
</table>

Table 8: Reward statistics for the same prompt across training steps. Human experts identified HPS-induced reward hacking beginning at step 200.

(a) Mean reward.

| Reward | 100 | 200 | 300 | 400 | 500 |
|---|---|---|---|---|---|
| ICT | 0.7835 | 0.8329 | 0.8729 | 0.8580 | 0.8658 |
| CLIP | 0.2879 | 0.2673 | 0.2200 | 0.1917 | 0.1592 |
| HPS | 0.2556 | 0.2528 | 0.2448 | 0.2296 | 0.2238 |
| HP | 0.7792 | 0.7796 | 0.7802 | 0.7799 | 0.7804 |

(b) Standard deviation.

| Reward | 100 | 200 | 300 | 400 | 500 |
|---|---|---|---|---|---|
| ICT | 0.1241 | 0.1180 | 0.0815 | 0.0857 | 0.0859 |
| CLIP | 0.0426 | 0.0416 | 0.0316 | 0.0245 | 0.0229 |
| HPS | 0.0079 | 0.0085 | 0.0070 | 0.0052 | 0.0064 |
| HP | 0.0011 | 0.0012 | 0.0007 | 0.0009 | 0.0007 |

(c) KL divergence.

| Reward | 100 | 200 | 300 | 400 | 500 |
|---|---|---|---|---|---|
| ICT | 5.9638 | **13.8767** | 5.9029 | 8.0059 | 9.6320 |
| CLIP | 5.5351 | **19.4165** | 4.9825 | 13.1837 | 18.0620 |
| HPS | 9.1868 | **19.9770** | 5.3309 | 13.0917 | 19.5950 |
| HP | 4.8112 | **14.1934** | 6.2217 | 11.0483 | 11.3300 |

### D.5 PRIOR DETECTION OF REWARD MODEL

We utilize the discrimination accuracy on high-quality triplet preference datasets (Pick-High Dataset and Pick-a-Pic Dataset), as shown in Table 9, as a prior detection method to early identify weak and strong reward models.

Table 9: Human preference prediction accuracy on high-quality image datasets.

| Reward | CLIP | HPS | ICT | HP |
|---|---|---|---|---|
| Human Preference Accuracy (%) | 60.30 | 72.88 | **87.58** | **88.47** |

As shown in Table 9, the strong reward models identified through our post-hoc detection achieve substantially higher accuracy in human preference prediction (ICT: 87.58%, HP: 88.47%) compared to weak reward models (CLIP: 60.30%, HPS: 72.88%).

### E ADDITIONAL QUANTITATIVE SUPPLEMENTARY EXPERIMENTS

**More Multi-Reward Optimization Baselines** We compare our OT-based method against three alternative multi-reward optimization baselines—weighted sum, separate constraints, and simple

mapping after Pareto frontier extraction—in Table 10. As shown in the results, our optimal transport framework consistently achieves higher performance and exhibits substantially lower susceptibility to reward hacking. In addition, we provide a clearer algorithmic explanation for why OT outperforms simple mapping. From an optimization perspective, the advantage of optimal transport lies in its holistic distribution-level mapping: OT transports dominated samples as a source *distribution* to the target distribution on the Pareto frontier, adaptively accounting for the geometric structure and density of the entire sample set. In contrast, simple mapping performs only point-wise projection and cannot capture distribution-level correspondences, which may result in imbalanced allocation or local clustering on the frontier. This distribution-level alignment makes OT inherently more robust and globally optimal across multiple reward dimensions.

Table 10: Ablation study comparing our OT-based method with alternative multi-reward optimization baselines on Parti-Prompts.

| Method | ICT ($\uparrow$) | HP ($\uparrow$) | CLIP ($\uparrow$) | HPS ($\uparrow$) | JDR$_2$ ($\uparrow$) | JDR$_4$ ($\uparrow$) | JCR$_4$ ($\downarrow$) |
|---|---|---|---|---|---|---|---|
| 2:3:2:3 | 50.80 | 56.43 | 46.51 | **86.03** | 28.31 | 13.42 | 2.57 |
| Sep-Const | 49.45 | 57.23 | 46.63 | 61.21 | 28.25 | 10.78 | 6.68 |
| Frontier-Mapp | **58.50** | 53.00 | **50.50** | 51.50 | 30.50 | 8.50 | 5.00 |
| **Ours** | 56.43 | **85.23** | 43.63 | 61.70 | **47.98** | **17.10** | **2.39** |

**Ablation Study on Entropy Regularization Strength.** We present an ablation study on the entropy regularization strength $\varepsilon$ in Table 11, demonstrating the robustness of our method across different hyperparameter settings.

Table 11: Ablation study on entropy regularization strength $\varepsilon$ in Optimal Transport.

| Strength | ICT (%)$\uparrow$ | HP (%)$\uparrow$ | CLIP (%)$\uparrow$ | HPS (%)$\uparrow$ | JDR$_2$ (%)$\uparrow$ | JDR$_4$ (%)$\uparrow$ | JCR$_4$ (%)$\downarrow$ |
|---|---|---|---|---|---|---|---|
| $\varepsilon = 0.1$ | 56.43 | 85.23 | 43.63 | 61.70 | 47.98 | 17.10 | 2.39 |
| $\varepsilon = 0.5$ | 57.50 | 80.41 | 51.70 | 69.00 | 49.20 | 16.50 | 3.50 |
| $\varepsilon = 0.9$ | 56.00 | 78.50 | 48.70 | 68.00 | 46.05 | 16.90 | 2.44 |

**Generalizability Across Post-Training Methods.** Our method is broadly applicable to any gradient-based post-training approach. We designed our framework to be training-algorithm-agnostic, requiring only that the base post-training method uses rewards to provide optimization signals. To demonstrate this generalizability, in addition to DRAFT-K presented in the main paper, we conduct additional experiments on ReFL Xu et al. (2023)in Table 12.

**Extension to 6 Reward Models and Computational Efficiency.** We scale from four to six reward models, with results shown in Table 13. Our method not only achieves strong performance but also maintains high computational efficiency.. The total time per iteration increases from 6.18s to 6.67s, representing only an 8% overhead. The OT loss computation increases from 161.17ms to 230.79ms, while component-wise overhead remains minimal: Pareto frontier extraction (0.44ms), optimal transport solving (2.1ms), and GPT-4o reward hacking detection (15s every 100 steps, amortizing to 0.15s per step). These results demonstrate that our framework scales efficiently with additional reward models, maintaining practical feasibility even with larger reward ensembles.

Table 12: Quantitative Results (%) on Parti-Prompts Dataset using ReFL.

| Model | ICT (%)$\uparrow$ | HP (%)$\uparrow$ | CLIP (%)$\uparrow$ | HPS (%)$\uparrow$ | JDR$_2$ (%)$\uparrow$ | JDR$_4$ (%)$\uparrow$ | JCR$_4$ (%)$\downarrow$ |
|---|---|---|---|---|---|---|---|
| Baseline 2:3:2:3 | 51.05 | 51.50 | **50.50** | 64.50 | 24.50 | 9.24 | 7.45 |
| **Ours** | **52.08** | **61.46** | 49.69 | **68.32** | **31.86** | **13.54** | **4.72** |

**Scalability to Multiple Reward Models.** In terms of computational efficiency, weighted sum methods face a combinatorial explosion problem: When optimizing $n$ rewards using weighted sum methods, if each weight is selected from $m$ candidate values, the computational cost of weight selection grows combinatorially, with grid search complexity of $O(m^n)$, rendering it computationally infeasible when $n$ reaches dozens or hundreds. In contrast, our method leverages the distinguishability of collapse signatures across different reward models, achieving $O(n)$ complexity for reward hacking

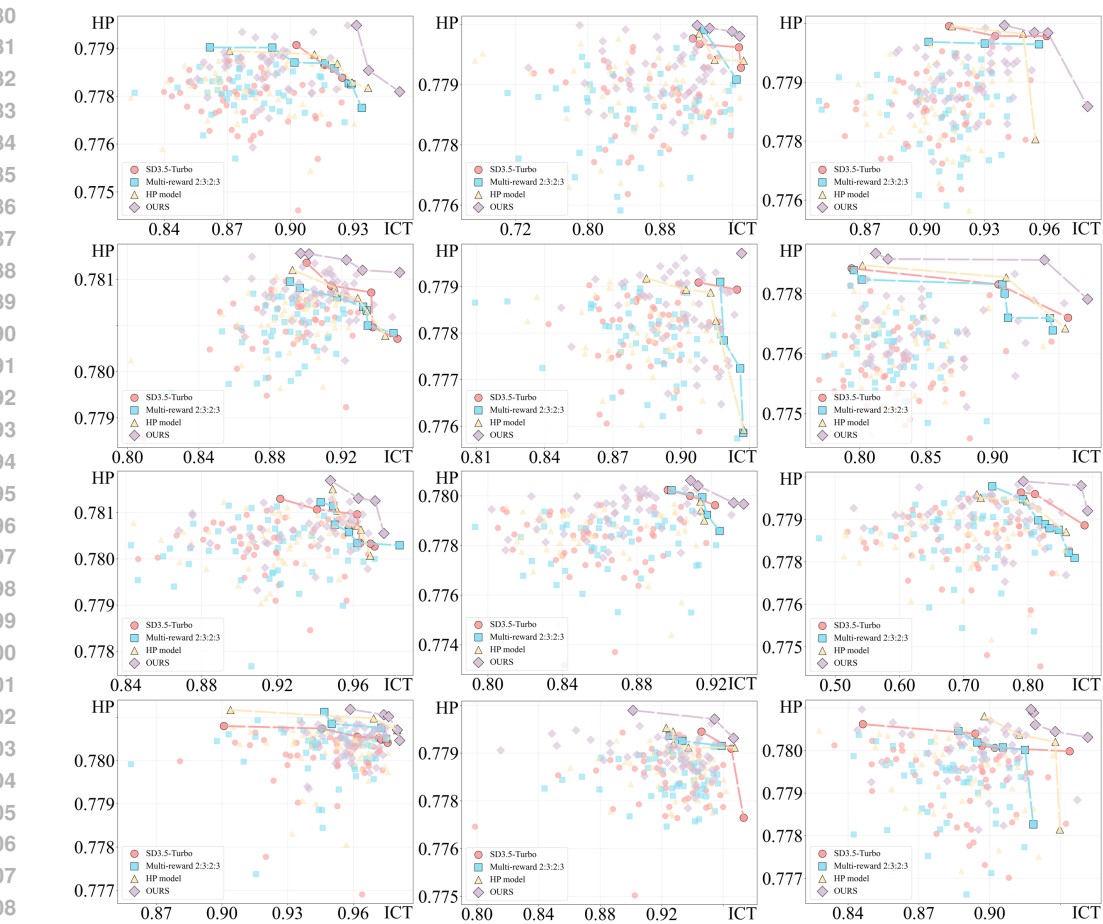

Figure 6: Broad Comparative Examples of Pareto Frontier Visualizations for Various Methods.

Table 13: Quantitative Results (%) on Parti-Prompts Dataset with 6 Reward Models.

| Model | ICT↑ | HP↑ | CLIP↑ | HPS↑ | JDR$_2$↑ | JDR$_4$↑ | JDR$_6$↑ | JCR$_4$↓ | JCR$_6$↓ |
|---|---|---|---|---|---|---|---|---|---|
| 6 rewards (Baseline) | 48.5 | 39.5 | **47.0** | 45.0 | 19.0 | 6.0 | 2.2 | 13.5 | 5.5 |
| 6 rewards (**Ours**) | **54.2** | **63.5** | 36.5 | **68.5** | **43.5** | **16.5** | **9.3** | **2.5** | **5.0** |

detection that scales linearly with the number of rewards. This linear scaling property makes our approach particularly advantageous for scenarios involving a large number of reward models, where traditional weighted sum approaches become prohibitively expensive.

## F  ADDITIONAL VISUALIZATION RESULTS

**Visualization and Analysis of Training Curve.** As shown in Figure 5, we present the joint domination rate JDR$_2$ on two strong rewards, ICT and HP. It can be observed that our method steadily improves as training progresses, whereas the baseline methods—whether single-reward optimization or multi-reward joint optimization—experience a rapid decline in joint domination rate, indicating that the baseline approaches quickly encounter reward hacking issues.

**Qualitative Case Studies on Pareto Frontier Visualization.** As shown in Figure 6, we provide additional visualization examples of Pareto frontiers. It can be observed that the Pareto frontier characteristics vary across different prompts. Our method consistently dominates the baseline approaches, and the overall image distributions are closer to the Pareto frontier, demonstrating the superiority of our method.

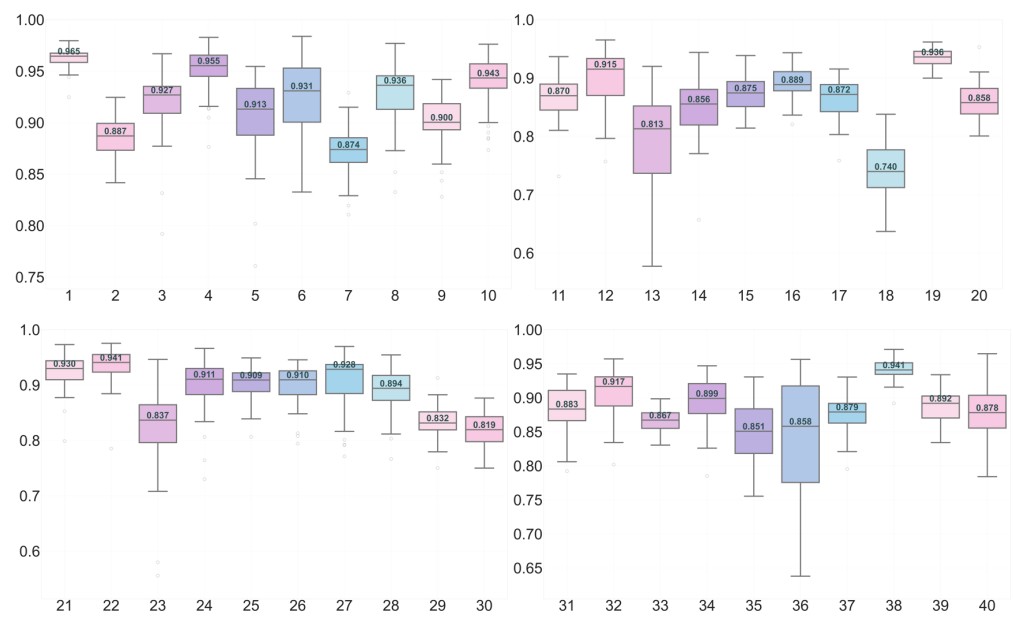

Figure 7: Visualization of Box Plots Showing Reward Variations Across Prompts on ICT Score.

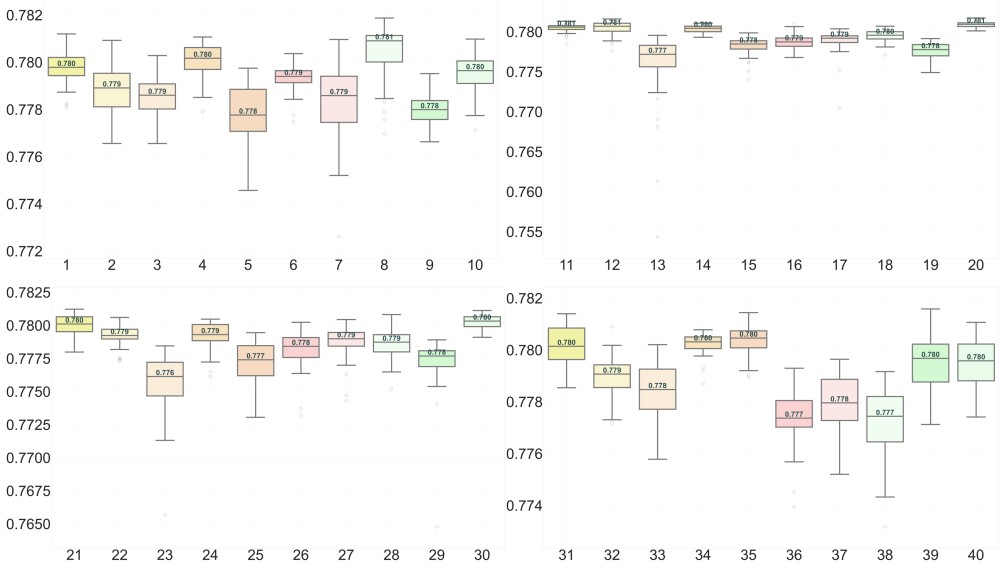

Figure 8: Visualization of Box Plots Showing Reward Variations Across Prompts on HP Score.

**Visualization of Heterogeneous Reward Bounds Across Different Prompts** As shown in 7, we provide box plot visualizations of reward ranges for the ICT score across different prompts, based on 50 images generated with different seeds. In 8, we present the corresponding box plot visualizations for the HP score across different prompts. It can be observed from these extensive examples that the reward upper bounds are heterogeneous across prompts, and that the reward ranges and characteristics also vary between prompts.

**More Qualitative Comparison Results.** We present additional visualizations for comprehensive comparison. Figure 9 shows results against single-reward baselines, while Figure 10 illustrates comparisons with multi-reward baselines.

## G  EXPERIMENT DETAILS

For diffusion model optimization, we use 32,000 non-repetitive text prompts from the Pick-High dataset, a subset of the Pick-a-pic dataset. All experiments are conducted on a cluster of three nodes, each with eight A800 GPUs. We adopt DDIM sampling with four denoising steps, set the classifier-free guidance weight to $0.0$, and fix the output resolution to $512 \times 512$. The Adam optimizer is used for training with a learning rate of $5 \times 10^{-5}$.

**Computational Overhead.** We provide a detailed analysis of our method's overhead in two parts: one-time precomputation and per-iteration overhead during training.

*Precomputation Cost.* The initial Pareto frontier is established by generating 50 images per prompt ($M = 50$). This one-time precomputation takes approximately 2 hours, accounting for about 12% of total training time. Notably, this initialization accelerates convergence by approximately 5×, making it a worthwhile investment that significantly enhances training efficiency.

*Per-Iteration Overhead.* During training, our method introduces minimal computational overhead compared to baseline methods. The runtime per iteration increases by only 3% (6.18s vs. 6.0s). The additional time is primarily attributed to optimal transport solving (1.8ms) and Pareto frontier extraction (0.4ms). The OT loss computation takes 161.17ms, only about 80ms more than the baseline's weighted sum (81.34ms)—a difference negligible compared to the dominant backward pass computation time. Additionally, GPT-4o reward hacking detection is performed once every 100 steps, taking 12 seconds per execution, which amortizes to 0.12s per step.

In summary, our method introduces modest computational overhead (2h one-time precomputation + 3% per-iteration increase) but delivers substantial practical benefits that baseline methods cannot achieve. While baselines typically suffer from rapid training collapse under identical settings, our method maintains stable training dynamics, achieves significantly superior performance, and effectively mitigates reward hacking issues—demonstrating that the computational investment is well justified by the qualitative improvements in training stability and final model quality.

**Reward Models and Training Strategy.** We employ four reward models, encompassing both strong and weak categories, to initialize joint training. These are grouped into two primary types: text–image alignment rewards (**CLIP** Radford et al. (2021) and **ICT** Ba et al. (2025) ) and human preference rewards (**HPS** Wu et al. (2023) and **HP** Ba et al. (2025)). Our staged optimization strategy proceeds in three phases. First, all four reward models are jointly optimized during the offline policy stage. Next, weak rewards that induce instability or collapse are adaptively pruned according to agent feedback. Finally, the remaining strong reward models (**ICT** and **HP**) are retained to guide the online policy stage for deeper optimization.

**Evaluation Metrics.** To comprehensively assess the effectiveness of multi-reward optimization, we adopt Joint Performance metrics as the primary evaluation criterion, including the Joint Domination Rate (**JDR**) and Joint Collapse Rate (**JCR**). Specifically, we report $\text{JDR}_2$, computed on the two strong rewards ICT and HP that are ultimately retained by the agent's decision-making process, as well as $\text{JDR}_4$ and $\text{JCR}_4$, both computed over all four rewards ICT, HP, CLIP, and HPS to assess the overall optimization capability. In addition, we provide the average scores from seven widely used reward models as supplementary evaluation to validate the robustness of our approach: Aesthetic Score[1] for aesthetic quality, CLIP Radford et al. (2021) for text–image consistency, ICT Ba et al. (2025) for the degree of text presence in images, and human preference models such as PickScore Kirstain et al. (2023), HPS Wu et al. (2023), ImageReward Xu et al. (2023), and HP Ba et al. (2025).

**Baseline Construction  Single-Reward Fine-Tuning.** Using Stable Diffusion 3.5-Turbo as the backbone, we fine-tune the model with CLIP, ICT, HPS, and HP as individual optimization objectives, employing DRaFT-K Clark et al. (2024). All experiments share identical hyperparameters, and for each reward, the best pre-collapse checkpoint verified by human experts is taken as the baseline.

**Weighted Multi-Reward Fine-Tuning.** Using the weighted loss in Equation 2, we jointly optimize the four rewards(CLIP, ICT, HPS, and HP) under identical settings, systematically exploring diverse weight combinations. The best pre-collapse checkpoint validated by experts is taken as the baseline.

---

[1]https://github.com/christophschuhmann/improved-aesthetic-predictor

**Reward Soup.** We adopt an inference-time fusion strategy, where the LoRA weights obtained from individual single-reward fine-tuned models are combined through weighted fusion. Specifically, this method dynamically merges parameters from multiple reward-specialized models during inference, thus exploring a broader reward fusion space without incurring additional training costs.

## H  THE USE OF LARGE LANGUAGE MODELS (LLMS)

In this study, Large Language Models (LLMs) were primarily used for language polishing and played a limited supporting role during the experimental process. Specifically, LLMs were employed to refine textual expressions, improve clarity, and assist with partial diagnostic analysis and strategy judgments in multi-reward model training. Beyond these limited supportive tasks, all experimental design, implementation, and result verification were carried out independently by the authors to ensure that the core ideas and scientific contributions of the research remain entirely author-driven.

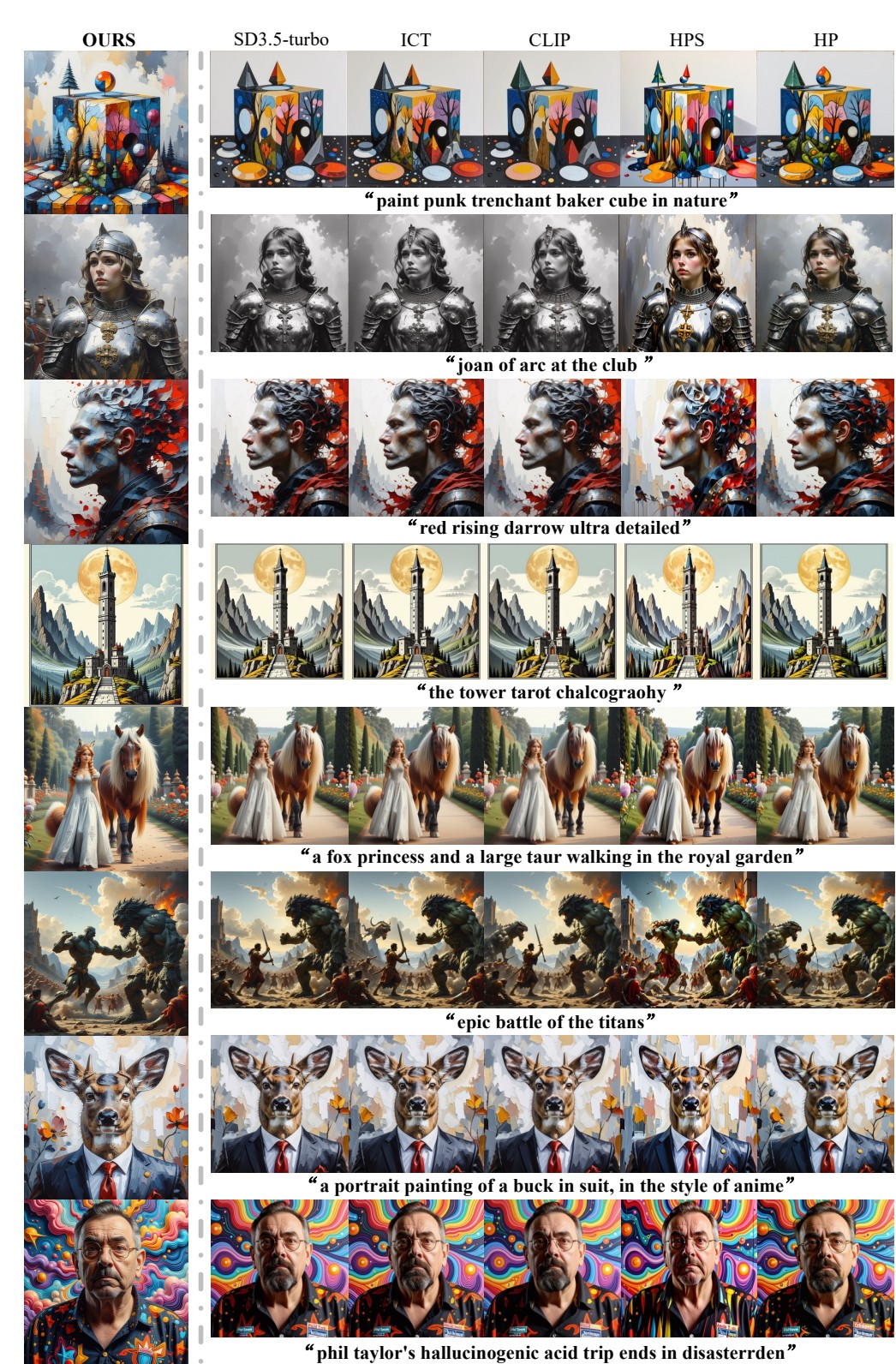

Figure 9: Qualitative Comparison of Optimization Results with Single-Reward Baselines.

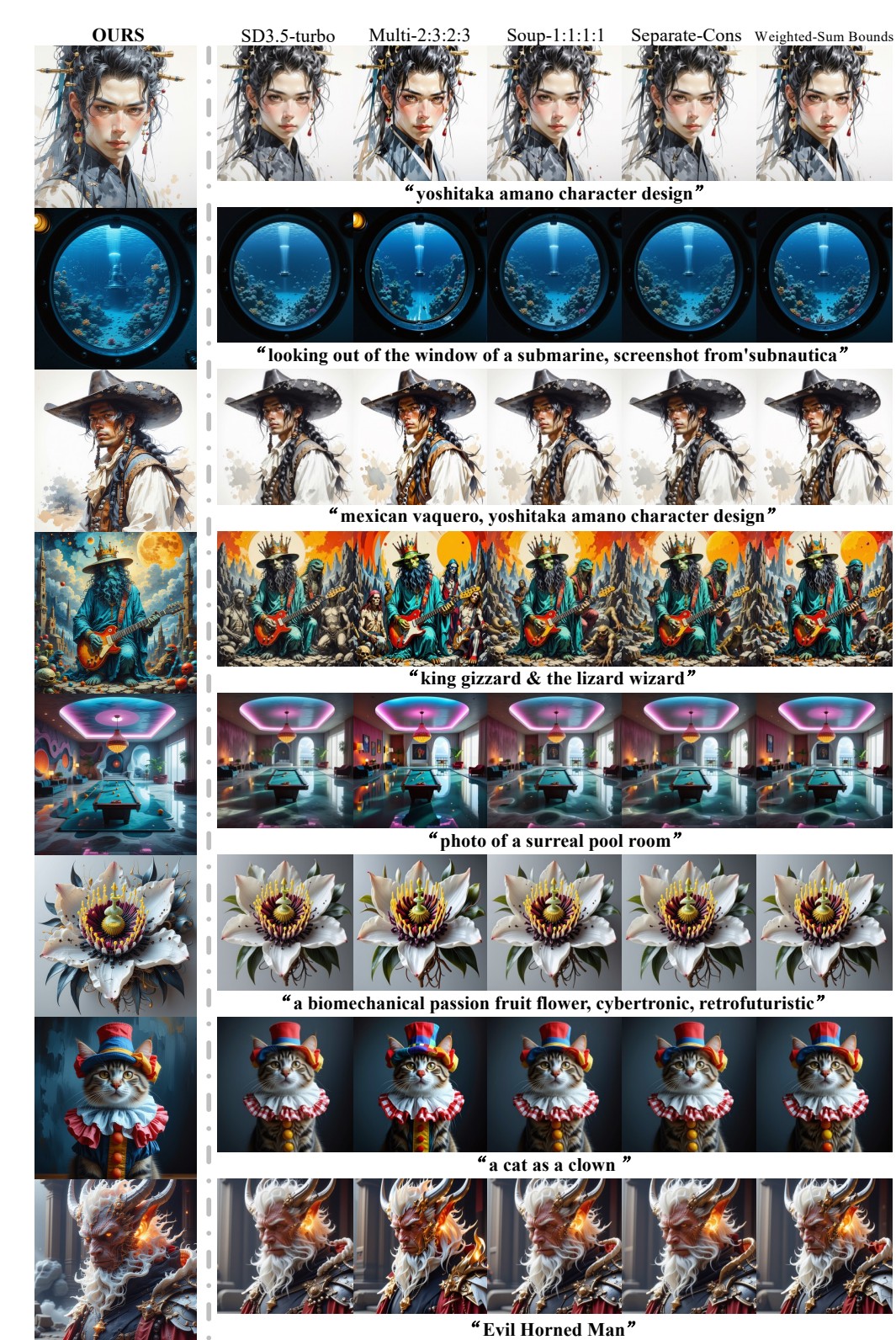

Figure 10: Qualitative Comparison of Optimization Results with Multi-Reward Baselines.

