# OpenReview forum: "Mitigating Reward Hacking in Multi-Reward Optimization for Text-to-Image Generation"
_ICLR.cc/2026/Conference — Submitted to ICLR 2026_

### Official Review · Reviewer_PiGz · 2025-10-27

**Soundness:** 3
**Presentation:** 2
**Contribution:** 3
**Rating:** 6
**Confidence:** 4

**Summary:**

The paper investigates reward hacking in multi-reward optimization for text-to-image models and argues that a single global bound across prompts can induce hacking because per-prompt reward ceilings are heterogeneous. It formalizes this observation with a property and a proposition, distinguishes strong and weak rewards, and proposes a Pareto-frontier-guided optimal transport framework. The framework has an offline stage that extracts prompt-wise Pareto fronts and uses an agent to prune weak rewards, followed by an online stage that moves samples toward dominating points on the fronts. The paper introduces Joint Domination Rate (JDR) and Joint Collapse Rate (JCR) to measure joint gains and collapses across rewards. Experiments on Parti-Prompts and a user study on DiffusionDB with SD3.5-Turbo plus LoRA and DRaFT-K report consistent improvements.

**Strengths:**

The paper presents a clear causal story from heterogeneous reward ceilings to reward hacking and supports it with formal definitions that motivate per-prompt targets. The Pareto-frontier plus optimal transport formulation is concrete and implementable, and the split into offline frontier extraction, agent-mediated pruning of weak rewards, and online frontier exploration is a practical recipe. JDR and JCR focus on per-sample, all-dimension consistency and align with the core thesis. The experiments cover single-reward, weighted multi-reward, reward-soup, and heterogeneous-bound baselines and the results show consistent gains with informative frontier visualizations.

**Weaknesses:**

1. **Positioning and ablation of the OT step:** The approach is conceptually related to Pareto-style multi-reward methods, but the paper does not isolate what the optimal transport mapping adds beyond non-dominated sorting or a simpler projection to the frontier. A small controlled study under matched base model, data, and token budget that compares a weighted sum, separate constraints, non-dominated sorting without OT, and the proposed OT loss would clarify the algorithmic delta and strengthen the claim of necessity.

2. **Operationalization of weak rewards and reliance on a proprietary agent:** The definitions of strong and weak rewards are motivated qualitatively, yet the paper does not report empirical correlations with human labels, threshold choices, or selection diagnostics. The pruning relies on a proprietary GPT-4o agent, which raises reproducibility concerns. Reporting reward–human correlations on a held-out labeled set, the decision thresholds with error analysis, and a simple non-proprietary fallback heuristic would make the procedure more transparent and reproducible.

**Questions:**

1. **Positioning and ablation of the OT step:** Run one minimal controlled replacement of the OT mapping with (i) non-dominated sorting + projection and (ii) a weighted-sum objective, under the same base model/data/token budget, and briefly explain what gradient signal OT introduces that the others do not? If adding runs is infeasible, a compact 2-D toy frontier with analytic gradients would still clarify when OT materially changes the solution.

2. **Weak-reward detection and reliance on a proprietary agent:** I highly recommend the authors to share a small diagnostic (e.g., ~200 held-out prompts) reporting each reward’s correlation with human labels and the pruning threshold you used, plus an ablation comparing “no pruning” vs “GPT-4o pruning” vs a simple open fallback (e.g., z-score filter + monotonicity checks)? Even approximate numbers would help assess false-pruning risk and reproducibility without proprietary tools.

---

> ### Author Response · Authors · 2025-11-26
> **Response to Reviewer PiGz (part 1)**
>
> Thank you for your comments and helpful suggestions. We address your concerns in detail below.
>
>
> **Q1:** More ablation studies on Optimal Transport？
>
>
> **A1:** We appreciate the reviewer's thoughtful recommendations. We provide experimental results for three baselines in Table 9: weighted sum, separate constraints, and simple mapping after finding the Pareto frontier using non-dominated sorting. As shown in Table 9, our optimal transport method achieves higher performance and is less prone to reward hacking.
>
>
> **Table 9: Ablation study comparing our OT-based method with alternative multi-reward optimization baselines on Parti-Prompts.**
>
> | Method | ICT↑ | HP↑ | CLIP↑ | HPS↑ | JDR₂↑ | JDR₄↑ | JCR₄↓ |
> |:-------|:----:|:---:|:-----:|:----:|:-----:|:-----:|:-----:|
> | Weighted Sum(Best,2:3:2:3) | 50.80 | 56.43 | 46.51 | **86.03** | 28.31 | 13.42 | 2.57 |
> | Separate Constraints | 49.45 | 57.23 | 46.63 | 61.21 | 28.25 | 10.78 | 6.68 |
> | Pareto frontier + Simple Mapping | **58.50** | 53.0 | **50.50** | 51.50 | 30.50 | 8.50 | 5.00 |
> | **Ours (OT-based)** |  56.43|**85.23** |43.63 | 61.70| **47.98** | **17.10** |**2.39** |
>
>
>
>
>
>
>
> From an algorithmic perspective, the advantage of optimal transport mapping over simple mapping lies in: OT transports dominated points as a source distribution holistically to the target distribution on the Pareto frontier, adaptively considering the geometric structure and density characteristics of the entire distribution; whereas simple mapping only performs point-wise projection and cannot capture distribution-level correspondences, potentially leading to imbalanced sample allocation or local clustering on the frontier. This distribution-level optimization makes the OT method more robust and globally optimal in terms of improvements across multiple reward dimensions.

---

> ### Author Response · Authors · 2025-11-26
> **Response to Reviewer PiGz (part 2)**
>
> **Q2:** Lack of  Reproducibility in Agent-Based Detection？
>
> **A2:** We provide detailed Agent design and workflow diagrams in Appendix C. We leverage the rich experience of human experts as strong prior knowledge for the Agent, rather than adopting simple threshold-based filtering schemes. To clearly demonstrate the reproducibility of our Agent, we present in Table 3 the agreement accuracy between its reward hacking detection and human expert assessments, where GPT-4o achieves an accuracy of 90.5%.
>
> Additionally, we provide experimental results on reward statistics (mean, variance, and KL divergence) in Table 5 for the same prompt across training steps, and apply z-score filtering combined with monotonicity checks to evaluate whether simple statistical methods can detect reward collapse. Human experts identified that reward collapse occurs at step 200, specifically caused by HPS reward hacking.
>
>
>
> Z-score filtering on the mean values (Table 5(a)) shows that HPS exhibits only a minor decrease at the collapse point (step 200: 0.2556→0.2528), failing to highlight it as anomalous through statistical thresholds.
>
> **Table 5: Reward statistics for the same prompt across training steps. Human experts identified reward collapse at step 200 caused by HPS.**
>
> **Table 5(a): Mean reward.**
>
> | Reward | 100    | 200    | 300    | 400    | 500    |
> |--------|--------|--------|--------|--------|--------|
> | ICT    | 0.7835 | 0.8329 | 0.8729 | 0.8580 | 0.8658 |
> | CLIP   | 0.2879 | 0.2673 | 0.2200 | 0.1917 | 0.1592 |
> | HPS    | 0.2556 | 0.2528 | 0.2448 | 0.2296 | 0.2238 |
> | HP     | 0.7792 | 0.7796 | 0.7802 | 0.7799 | 0.7804 |
>
> Similarly, variance analysis (Table 5(b)) reveals that all rewards maintain relatively stable standard deviations across steps, with no distinctive pattern at step 200 to indicate HPS-specific issues.
> **Table 5(b): Standard deviation.**
>
> | Reward | 100    | 200    | 300    | 400    | 500    |
> |--------|--------|--------|--------|--------|--------|
> | ICT    | 0.1241 | 0.1180 | 0.0815 | 0.0857 | 0.0859 |
> | CLIP   | 0.0426 | 0.0416 | 0.0316 | 0.0245 | 0.0229 |
> | HPS    | 0.0079 | 0.0085 | 0.0070 | 0.0052 | 0.0064 |
> | HP     | 0.0011 | 0.0012 | 0.0007 | 0.0009 | 0.0007 |
>
>
> When applying z-score filtering to KL divergence (Table 5(c)), all rewards exhibit high values at step 200 (ICT: 13.88, CLIP: 19.42, HPS: 19.98, HP: 14.19), with similar anomalous patterns appearing at other steps (e.g., step 500). Monotonicity checks further reveal that all rewards display irregular trends without clear patterns, making it impossible for simple statistical methods to disentangle which specific weak reward caused the collapse. This is because when collapse occurs, all reward distributions undergo dramatic shifts simultaneously due to the degraded image quality.
>
> ​**Table 5(c): KL divergence.**
>
>
> | Reward | 100      | **200**    | 300      | 400      | 500      |
> |--------|----------|----------|----------|----------|----------|
> | ICT    | 5.9638   | **13.8767**  | 5.9029   | 8.0059   | 9.6320 |
> | CLIP   | 5.5351   | **19.4165**  | 4.9825   | 13.1837  | 18.062 |
> | HPS    | 9.1868   | **19.9770**  | 5.3309   | 13.0917  | 19.595 |
> | HP     | 4.8112   | **14.1934**  | 6.2217   | 11.0483  | 11.330 |
>
>
> These results demonstrate that while simple statistical heuristics are insufficient for reliable weak reward detection, our GPT-4o-based Agent achieves 90.5% agreement with human experts, providing a reproducible and transparent solution grounded in domain knowledge.

---

> > ### Comment · Reviewer_PiGz · 2025-11-26
> >
> > Thank you for the detailed and well-structured rebuttal. It has addressed my main concerns and clarified the issues I raised. I am satisfied with the authors’ responses and will keep my positive rating for this paper.

---

### Official Review · Reviewer_aapt · 2025-10-31

**Soundness:** 3
**Presentation:** 4
**Contribution:** 3
**Rating:** 4
**Confidence:** 3

**Summary:**

The paper introduces an innovative approach to optimizing multi-reward models for text-to-image generation, employing Pareto frontiers and optimal transport theory to guide solutions towards Pareto-optimality, significantly enhancing model performance. It features a GPT-4o decision agent for adaptive multi-reward model training, identifying and managing weak reward models prone to reward hacking. Enhanced metrics JDR and JCR are introduced for reliable multi-reward optimization assessment. Experimental results demonstrate superior efficacy in improving text-to-image models and avoiding reward manipulation compared to strong baselines.

**Strengths:**

1.	The paper offers a compelling insight of reward hacking: the conflict between a "unified global optimization target" and the "heterogeneity of rewards" as the core issue is a compelling insight.
2.	This paper introduces a novel method using Pareto frontiers and optimal transport theory to optimize multi-reward models in text-to-image generation tasks. This approach potentially mitigate the reward hacking issue by guiding the optimization towards Pareto-optimal solutions.
3.	As performance metrics specifically designed for multi-reward optimization, the Joint Dominance Ratio (JDR) and Joint Collapse Ratio (JCR) provide a more precise way to evaluate the effectiveness of different training strategies.

**Weaknesses:**

1.	The proposed method's complexity involving: 1) pre-computing a candidate set of samples, 2) extracting the Pareto frontier, 3) solving an optimal transport problem, and 4) employing a GPT-4o agent for monitoring. This introduces significant computational overhead and implementation challenges.
2.	The paper does not thoroughly examine whether the proposed method generalizes to other text-to-image models or scales with an increased number of reward models. It is not clear how this method will perform in more complex scenarios.

**Questions:**

1.	Could the authors elaborate on the computational cost of the proposed method? Specifically, how much does it increase training time and GPU resource consumption compared to the baselines? Is there a possibility for a "lightweight" implementation that could significantly reduce complexity and cost at the expense of some performance?
2.	While the proposed method has shown excellent results with four reward models. How do the authors foresee its performance when using a much larger set of reward models? At that scale, would the computation of the Pareto frontier and the solving of the optimal transport problem become bottlenecks?
3.	The classification of reward models is vital to the proposed strategy. An expert-driven decision requiring significant prior knowledge, could the authors provide more details regarding its robustness?

---

> ### Author Response · Authors · 2025-11-26
> **Response to Reviewer aapt**
>
> Thank you for your comments and attention to the computational details and generalization. We address your concerns in detail below.
>
> **Q1:** The complexity of the proposed method.
>
>
> **A1:** We sincerely appreciate the reviewer's thoughtful consideration of our method. We understand the concern regarding method complexity, but we would like to clarify an important evaluation perspective:
>
> Post-training aims to achieve better performance while preventing reward hacking, not minimizing training time. Under the same experimental settings, baseline methods typically collapse quickly and achieve significantly weaker metrics than our method. In contrast, our method can stably improve performance and effectively mitigate reward hacking(**Section 5 Figure 3**). To clearly demonstrate the computational overhead, we provide a detailed comparison of runtime costs:
>
>
> **Pareto Frontier Precomputation (one-time, before training): 2h**
>    - 12% of total training time
>    - **Key benefit:** Provides clear optimization targets from the start, enabling 5× faster convergence
>
> **Per-Iteration Runtime:**
> - **Total time:** Baseline (6.0s) vs. Ours (6.18s) — **3% overhead only**
> - **Loss computation:** Baseline weighted sum (81.34ms) vs. Ours OT loss (161.17ms)
>   - *Note: The 80ms difference is negligible as the backward pass dominates most of the iteration time*
>
> **Component-wise Overhead per Iteration (Ours):**
> - Pareto frontier extraction: 0.4ms
> - Optimal transport solving: 1.8ms
> - GPT-4o reward hacking detection: 12s every 100 steps (amortized: 0.12s/step)
>
>
> **Summary:** Our method adds minimal overhead (2h precomputation + 0.18s per iteration + 0.12s/step for GPT-4o detection) but delivers substantially better performance with stable training and effective reward hacking prevention, which baselines cannot achieve. Notably, the 2h precomputation accelerates our own convergence by 5× (compared to training without precomputation), making it a worthwhile one-time investment.
>
>
> **Q2:** Performance and computational bottlenecks when using a larger set of reward models
>
> **A2:** We sincerely appreciate the reviewers' concern regarding scalability. To address this, we conducted additional experiments incorporating two more reward models (PickScore and ImageReward), expanding our framework to six reward models. Our analysis demonstrates that the computational overhead remains manageable and does not constitute a bottleneck.
>
> **Computational Efficiency Analysis:**
> - Total time per iteration: 4 rewards (6.18s) → 6 rewards (6.67s), representing only an 8% increase
> - OT loss computation: 4 rewards (161.17ms) → 6 rewards (230.79ms)
> - Component-wise overhead per iteration (6 rewards):
>   - Pareto frontier extraction: 0.44ms
>   - Optimal transport solving: 2.1ms
>   - GPT-4o reward hacking detection: 15s every 100 steps (amortized to 0.15s per step)
>
> These results indicate that our framework scales efficiently with additional reward models.
>
> **Generalization Performance:**
> As shown in Table 8, our method maintains strong performance and generalization capability even when scaling to six reward models. The results validate that our approach effectively handles larger reward model ensembles without compromising either efficiency or effectiveness.
>
> **Table 8: Quantitative Results (%) on Parti-Prompts Dataset with 6 Reward Models.**
>
> | Model | ICT (%)↑ | HP (%)↑ | CLIP (%)↑ | HPS (%)↑ | JDR₂ (%)↑ | JDR₄ (%)↑ | JDR₆ (%)↑ | JCR₄ (%)↓ | JCR₆ (%)↓ |
> |:------|:--------:|:-------:|:---------:|:--------:|:---------:|:---------:|:---------:|:---------:|:---------:|
> | 6 rewards (Baseline) | 48.5 | 39.5 | **47.0** | 45.0 | 19.0 | 6.0 | 2.2 | 13.5 | 5.5 |
> | 6 rewards (**Ours**) | **54.2** | **63.5** | 36.5 | **68.5** | **43.5** | **16.5** | **9.3** | **2.5** | **5.0** |
>
>
>
>
> **Q3:**  More details on Agent robustness？
>
> **A3:** We provide detailed Agent design details in Appendix C, including using single-reward optimization to generate the model to obtain distinguishable collapsed images and corresponding feature descriptions, etc. To verify the robustness of our Agent, we demonstrate in Table 3 that using 200 different collapsed images, we obtained a 90.5% accuracy correlation between Agent detection and human expert decisions, demonstrating that the Agent can perform accurate and stable evaluation when equipped with strong human expert-guided prior knowledge.

---

### Official Review · Reviewer_Xf1N · 2025-10-31

**Soundness:** 2
**Presentation:** 2
**Contribution:** 2
**Rating:** 4
**Confidence:** 3

**Summary:**

This paper tackles the problem of reward hacking in multi-reward optimization for T2I models. The authors identify a fundamental flaw in existing methods: they use a unified global optimization target (a single upper bound for rewards), which fails to account for the fact that different text prompts have heterogeneous reward landscapes and achievable upper bounds.

**Strengths:**

Pros:
1. The identification of "heterogeneous reward bounds vs. a unified global target" as the root cause of reward hacking is an interesting insight. It provides a theoretical foundation for the proposed solution.
2. The use of Pareto frontiers and Optimal Transport is a natural and mathematically sound approach for multi-objective optimization. It directly addresses the core problem of prompt-specific targets in a structured way.
3. The evaluation is comprehensive. It includes a strong set of baselines, robust quantitative analysis using both standard and their new metrics, compelling qualitative examples, a human user study, and a detailed ablation study that validates each component of their framework.

**Weaknesses:**

Cons:
1. The use of GPT-4o as a core component in the training loop is a potential weakness:
    1. It makes the method less reproducible and accessible, as it relies on a costly, black-box external API whose behavior could change over time.
    2. The feasibility of this approach for dozens or hundreds of rewards is questionable.
    3. The paper would be strengthened by exploring or discussing simpler, fully automatable heuristics for detecting reward collapse (e.g., based on reward score variance, KL divergence from a reference distribution, or simpler image artifact detectors) as alternatives to the LLM agent.
    4. Have you explored replacing the GPT-4o agent with a more traditional, programmatic method for detecting reward collapse? For instance, could you monitor the distribution of reward scores and flag a model for pruning if its variance spikes or its mean diverges too quickly from a moving average?
2. The formal definitions rely on the correlation with an unobserved "human-perceived quality Q". In practice, the classification seems to be done post-hoc by the GPT-4o agent observing collapse during training. The link between the formal definition and the practical implementation could be tighter. An a priori method for estimating reward model strength would make the framework more efficient.
3. The paper mentions using the Sinkhorn algorithm but does not discuss the choice of hyperparameters (e.g., entropy regularization strength ε) or the sensitivity of the results to these choices.
4. Could you provide details on the scale of the offline precomputation? Specifically, how many samples per prompt (M in the paper) were needed to form a robust initial Pareto frontier, and how much did this contribute to the overall training time?
5.  In the online phase, how are new points added to the Pareto frontier? Is the entire frontier for a prompt re-calculated at each step using all historical and newly generated samples, or is there a more efficient updating mechanism?
6. How does the method perform on prompts that are stylistically or semantically very different from those used to build the initial offline Pareto frontiers? Does the online phase sufficiently adapt to out-of-distribution prompts?
7. The OT framework maps dominated samples to the frontier. How does the framework handle samples that are already very good (i.e., close to or on the frontier)? Is the loss signal for these samples near zero, effectively leaving them unchanged?

**Questions:**

See Weaknesses

---

> ### Author Response · Authors · 2025-11-26
> **Response to Reviewer Xf1N (part 1)**
>
> Thank you for your comments and helpful suggestions. We address your concerns in detail below.
>
> **Q1:** Questions of GPT-4o Agent.
>
>
> **Q1.1:** Does using GPT-4o as an agent face issues with reproducibility, accessibility, and high costs?
>
> **A1.1:** We thank the reviewer for their attention to reproducibility, accessibility, and cost concerns. We address each of these three aspects with explanations and strong validation experiments.
>
> **（1）Knowledge-driven agent design enhances reproducibility: 90.5% agreement with human experts**. Rather than using GPT-4o as a black-box detector for reward hacking, we design the agent with extensive prior knowledge grounded in human expertise (detailed in Appendix C), including: carefully curated examples of subtle hacking patterns across different rewards, reward hacking characteristics for different rewards extracted through GPT-4o's reasoning capabilities, and chain-of-thought planning. This design ensures that GPT-4o serves as a knowledge-enhanced reasoning agent rather than a black-box oracle. The rich human expert knowledge embedded in the agent's inputs provides a stable prior foundation, thereby mitigating reproducibility concerns. Furthermore, Table 3 shows that GPT-4o's reward hacking detection achieves 90.5% agreement with human expert assessments.
>
> **（2）Supplementary validation with open-source VLMs addresses accessibility.** While we use GPT-4o for its strong reasoning and visual understanding capabilities, our framework is model-agnostic. The agent component can be replaced with other vision-language models (e.g., Qwen3-VL-32B-Thinking[1], Qwen3-VL-8B-Thinking[1], GLM-4.5V[2], etc.) without fundamental changes to the methodology. Therefore, we additionally evaluate these three strong open-source models in Table 3, presenting the Agreement accuracy between their reward hacking detection and human expert assessments. As shown in the table, open-source models can achieve 87% accuracy, which is sufficient for detecting weak rewards during training when used as agents, without causing performance degradation.
>
> **Table 3: Agreement accuracy between VLM-based reward hacking detection and human expert assessments across 200 evaluation samples.**
>
> | Model | GPT-4o | Qwen3-VL-32B-Thinking[1] | GLM-4.5V[2] | Qwen3-VL-8B-Thinking[1]
> |-------|--------|----------|----------|---------|
> | Agreement Accuracy| 90.5% | 87.5%| 84%  | 83.5%| |
>
>
> **（3）Cost-effective API call: only 0.4% of training overhead.** We clarify in Appendix C that agent-based detection is periodically invoked at fixed training intervals, rather than at every optimization step. This sampling strategy significantly reduces the number of API calls while maintaining effective monitoring for reward hacking. A single complete agent-detection run costs approximately $0.015, and throughout the full training process, the cost of agent detection accounts for only about 0.4% of the total training expenditure.
>
>
>
>
> **Q1.2:** Concerns about scaling to multiple rewards.
>
> **A1.2:** In Appendix C, we demonstrate through single-reward experiments that different reward models exhibit distinguishable collapse patterns due to their unique training data and optimization objectives. By incorporating rich human expert knowledge as the agent's prior, the GPT-4o agent can accurately identify the specific collapse patterns induced by each reward model. We further conducted additional single-reward experiments on ImageReward and PickScore, observing that they similarly exhibit unique collapse patterns that the agent can reliably distinguish.
> In terms of computational efficiency, weighted sum methods face a combinatorial explosion problem: When optimizing n rewards using weighted sum methods, if each weight is selected from m candidate values, the computational cost of weight selection grows combinatorially, with grid search complexity of $O(m^n)$, rendering it computationally infeasible when n reaches dozens or hundreds. Our method leverages the distinguishability of collapse signatures across different reward models, achieving $O(n)$ complexity for reward hacking detection that scales linearly with the number of rewards.

---

> ### Author Response · Authors · 2025-11-26
> **Response to Reviewer Xf1N (part 2)**
>
> **Q1.3:** Additional suggestions on heuristic methods for detecting reward hacking.
>
>
> **A1.3:** We thank the reviewer for this valuable suggestion. We conducted comprehensive experiments to evaluate whether heuristic methods could effectively detect reward hacking. Specifically, we analyzed three common statistical indicators: mean reward, reward variance (std), and KL divergence from a reference distribution. We evaluated these metrics under two scenarios: (1) images generated from different prompts at the same training step (Table 4), and (2) images generated from the same prompt across different training steps (Table 5).
>
> Our results demonstrate that **no reliable heuristic pattern emerges** from these statistics:
>
> **（1）Cross-prompt heterogeneity (Table 4):** We generated 50 images per prompt at step 300 (where reward hacking has occurred) and found that different prompts exhibit drastically different reward statistics, including heterogeneous reward upper bounds and value ranges (**Appendix D, Figures 7-8**). Since each training step samples different prompts, these statistical measures are fundamentally unstable and cannot serve as a basis for effective heuristic detection.
>
>
> **Table 4: Reward statistics exhibit heterogeneity across different prompts.**
>
> **Table 4(a): Mean reward.**
> | Reward | Prompt 1 | Prompt 2 | Prompt 3 |
> |--------|----------|----------|----------|
> | ICT    | 0.8676   | 0.8114   | 0.9095   |
> | CLIP   | 0.2205   | 0.1421   | 0.2956   |
> | HPS    | 0.2449   | 0.2661   | 0.2749   |
> | HP     | 0.7800   | 0.7805   | 0.7805   |
>
>
> ​**Table 4(b): Standard deviation.**
> | Reward | Prompt 1 | Prompt 2 | Prompt 3 |
> |--------|----------|----------|----------|
> | ICT    | 0.09     | 0.09     | 0.06     |
> | CLIP   | 0.03     | 0.04     | 0.03     |
> | HPS    | 0.007    | 0.01     | 0.007    |
> | HP     | 0.0006   | 0.0009   | 0.0005   |
>
> ​**Table 4(c): KL divergence.**
>
> | Reward | Prompt 1 | Prompt 2 | Prompt 3 |
> |--------|----------|----------|----------|
> | ICT    | 8.00     | 5.45     | 11.42    |
> | CLIP   | 13.18    | 19.48    | 4.89     |
> | HPS    | 13.09    | 10.05    | 14.34    |
> | HP     | 11.04    | 10.78    | 16.82    |
>
>
> **（2）Reward indistinguishability (Table 5):** We measured reward statistics on images generated from the same prompt across different training steps (50 images per prompt), with human expert evaluation showing reward hacking caused by HPS beginning at step 200. The results reveal critical limitations of heuristic methods:
>
>
> - **Mean and standard deviation fail to detect anomalies (Tables 5(a) and 5(b)):** Neither metric exhibits clear abnormal patterns at the collapse point, making them unreliable for detecting reward hacking.
>
> - **KL divergence detects collapse but cannot disentangle rewards (Tables 5(c)):** While KL divergence shows sharp spikes that could signal reward hacking, all rewards exhibit similar sudden changes simultaneously. This makes it impossible to identify which specific reward is responsible for the collapse.
>
>
> **Table 5: Reward statistics for the same prompt across training steps.**
>
> **Table 5(a): Mean reward.**
>
> | Reward | 100    | 200    | 300    | 400    | 500    |
> |--------|--------|--------|--------|--------|--------|
> | ICT    | 0.7835 | 0.8329 | 0.8729 | 0.8580 | 0.8658 |
> | CLIP   | 0.2879 | 0.2673 | 0.2200 | 0.1917 | 0.1592 |
> | HPS    | 0.2556 | 0.2528 | 0.2448 | 0.2296 | 0.2238 |
> | HP     | 0.7792 | 0.7796 | 0.7802 | 0.7799 | 0.7804 |
>
> **Table 5(b): Standard deviation.**
>
> | Reward | 100    | 200    | 300    | 400    | 500    |
> |--------|--------|--------|--------|--------|--------|
> | ICT    | 0.1241 | 0.1180 | 0.0815 | 0.0857 | 0.0859 |
> | CLIP   | 0.0426 | 0.0416 | 0.0316 | 0.0245 | 0.0229 |
> | HPS    | 0.0079 | 0.0085 | 0.0070 | 0.0052 | 0.0064 |
> | HP     | 0.0011 | 0.0012 | 0.0007 | 0.0009 | 0.0007 |
>
>
>
> ​**Table 5(c): KL divergence.**
>
>
> | Reward | 100      | **200**    | 300      | 400      | 500      |
> |--------|----------|----------|----------|----------|----------|
> | ICT    | 5.9638   | **13.8767**  | 5.9029   | 8.0059   | 9.6320   |
> | CLIP   | 5.5351   | **19.4165**  | 4.9825   | 13.1837  | 18.062 |
> | HPS    | 9.1868   | **19.9770**  | 5.3309   | 13.0917  | 19.595 |
> | HP     | 4.8112   | **14.1934**  | 6.2217   | 11.0483  | 11.330 |
>
>
> In contrast, our knowledge-driven agent successfully addresses both challenges: detecting when reward hacking occurs and identifying which reward causes it. These comprehensive experiments confirm that simple heuristic methods based on reward statistics are insufficient for detecting reward hacking in text-to-image generation, thereby motivating and validating our agent-based approach.

---

> ### Author Response · Authors · 2025-11-26
> **Response to Reviewer Xf1N (part 3)**
>
> **Q2:** The connection between the defined "human-perceived quality Q" and the agent's post-hoc judgments during training  could be tighter.
>
>
> **A2:** We sincerely thank the reviewers for their insightful observations and valuable suggestions. Indeed, existing models are currently unable to directly output "human-perceived quality Q." However, the Agent we employ integrates prior knowledge from human experts and can serve as an effective proxy for human-perceived quality, thereby enabling the identification of image samples with declining perceived quality during the training process. Even if we can predefine Strong reward models and Weak reward models based on their definitions, the number of steps or duration that different reward models can sustain during training still needs to be determined through actual training. Therefore, we adopted a "post-hoc detection" strategy. However, we adopte the reviewers' suggestion and utilize the discrimination accuracy on high-quality triplet preference datasets(Pick-High Dataset and Pick-a-pic Dataset) in Table 6 as a prior detection method to identify and prevent reward model vulnerabilities at an earlier stage.
>
> As shown in Table 6, the strong reward models identified through our post-hoc detection achieve substantially higher accuracy in human preference prediction (ICT: 87.58%, HP: 88.47%) compared to weak reward models (CLIP: 60.30%, HPS: 72.88%). This significant performance gap validates that our approach effectively distinguishes between reliable and vulnerable reward models, further confirming the effectiveness of our detection framework.
>
> **Table 6: Human preference prediction accuracy on high-quality image datasets.**
> | Reward    | CLIP  | HPS   | ICT    | HP     |
> |--------------------|-------|-------|--------|--------|
> | Accuracy (%) | 60.30 | 72.88 | **87.58** | **88.47** |
>
>
>
>
> **Q3:** Sensitivity analysis of the entropy regularization strength ε in the Sinkhorn algorithm.
>
> **A3:** We present an ablation study on the entropy regularization strength ε in Table 7, demonstrating the robustness of our method.
>
>
> **Table 7: Ablation study on the entropy regularization strength ε.**
>
> | Regularization Strength | ICT (%)↑ | HP (%)↑ | CLIP (%)↑ | HPS (%)↑ | JDR₂ (%)↑ | JDR₄ (%)↑ | JCR₄ (%)↓ |
> |:------|:--------:|:-------:|:---------:|:--------:|:---------:|:---------:|:---------:|
> | ε = 0.1 | 56.43|85.23 |43.63 | 61.70| 47.98 | 17.10 |2.39 |
> | ε = 0.5 |57.50 | 80.41| 51.70| 69.00| 49.20|16.50 | 3.50 |
> | ε = 0.9 |56.00 | 78.50| 48.70| 68.00| 46.05|16.90 | 2.44 |
>
>
>
>
>
>
>
>
>
> **Q4:** Details of offline precomputation？
>
> **A4:** The initial Pareto frontier is established by generating 50 images per prompt (M=50). This precomputation, which is performed prior to training and takes approximately 2 hours, accounts for about 12% of the total training time. The precomputed data is then utilized directly during training. It is important to note that without this initialization, convergence would be approximately five times slower. Thus, the precomputation is an acceptable one-time cost that significantly enhances overall training efficiency.
>
>
> **Q5:** Construction of the Pareto frontier in the online phase？
>
> **A5:** We specify in Section 3.4 that the Pareto frontier is constructed by gathering generated samples from all processes and identifying the Pareto-optimal points. In this phase, the precomputed Pareto frontier is not used; instead, the frontier is derived entirely from the set of generated images themselves.
>
> **Q6:** Does it adapt to out-of-distribution prompts?
>
> **A6:** All our qualitative and quantitative experiments were validated on out-of-distribution (OOD) prompt datasets, including the Parti-Prompts dataset (**Section 5, Table 1 and Table 2**) and the DiffusionDB dataset (**Section 5, Table 3**). Extensive experiments demonstrate the effectiveness of our method in generalizing to OOD prompts.
>
> **Q7:** How are samples ​that are already very good​ handled?
>
> **A7:** For points that are sufficiently good or already on the Pareto frontier, we do not perform further optimization. This is because reward hacking occurs when the reward exceeds the theoretical optimum while actual human perception degrades. Therefore, we avoid further optimization of samples that are already near the theoretical optimum to prevent reward hacking.
>
>
>
>
>
> **References**
>
>
> [1] Yang, A., Li, A., et al. "Qwen3 Technical Report." arXiv preprint arXiv:2505.09388 (2025).
>
>
> [2]  V Team, Wenyi Hong, et al. "GLM-4.5V and GLM-4.1V-Thinking: Towards Versatile Multimodal Reasoning with Scalable Reinforcement Learning." arXiv preprint arXiv:2507.01006 (2025).

---

### Official Review · Reviewer_Wabn · 2025-11-01

**Soundness:** 3
**Presentation:** 3
**Contribution:** 3
**Rating:** 4
**Confidence:** 3

**Summary:**

This paper addresses reward hacking in multi-reward optimization for text-to-image generation. It proposes a Pareto frontier-guided optimal transport framework with prompt-specific frontiers and online/offline strategies tailored to reward model strengths. Two new metrics, Joint Domination Rate (JDR) and Joint Collapse Rate (JCR), are introduced for better evaluation. Experimental results show a 10% performance improvement over baselines, effectively mitigating reward hacking while enhancing multi-reward alignment.

**Strengths:**

1. This paper is well written and easy to follow
2. This article provides sufficient theoretical analysis of reward hacking and has a clear standpoint.

**Weaknesses:**

1. The Pareto optimal problem is used to address situations where there are conflicts between rewards. However, in the paper, there does not appear to be a clear conflict between the two types of rewards selected. Empirically, training with text-image alignment reward tends to lead to an improvement in human preference reward, and vice versa. It is suggested that the authors conduct experiments to investigate whether the proposed method remains effective when conflicts between rewards exist. For example, as mentioned in the paper, rewards related to authenticity and artistic style may conflict.

2. Is this method applicable to any arbitrary text-to-image post-training approach? For instance, methods like ReFL, FlowGRPO, etc. Can the authors provide experiments to validate and illustrate this?

**Questions:**

Please see the questions in the Weaknesses. I hope these issues can be resolved, and I will reconsider my grading.

---

> ### Author Response · Authors · 2025-11-26
> **Response to Reviewer Wabn**
>
> Thank you for your comments and thoughtful consideration of our work. We address your concerns in detail below.
>
>
> **Q1:** The two types of rewards selected in the paper do not appear to have clear conflicts, as they tend to improve simultaneously during training?
>
> **A1:** We thank the reviewer for their thoughtful consideration of reward conflicts in Pareto optimization. Indeed, rewards can improve simultaneously in suboptimal regions; however, this does not contradict the existence of conflicts. In multi-objective optimization, "conflict" refers to the trade-offs at the Pareto frontier, not complete opposition across the entire space. To alleviate the reviewer's confusion about the meaning of conflict, we provide here a detailed explanation of multi-reward conflicts and present experimental data to demonstrate that the rewards we use do indeed exhibit conflicts.
>
> 1.  **Detailed Explanation:** In Pareto multi-objective optimization, conflict does not mean that objectives are completely negatively correlated across the entire space, but rather refers to the trade-offs that exist at the Pareto frontier. According to the definition of Pareto optimality, a solution is Pareto optimal when no objective can be further improved without degrading at least one other objective(**Section 3.1**). This implies that multiple objectives can improve jointly in suboptimal regions, while trade-offs necessarily exist at the frontier.
> The phenomenon observed by the reviewer, that text-image alignment reward and human preference reward can mutually reinforce each other, occurs precisely in the early suboptimal phase of optimization. When one reward approaches optimality, further improvement inevitably requires sacrificing other reward dimensions, which is the essence of conflict. We present the two-dimensional Pareto frontier of ICT and HP (**Figure 2, Section 5**), where the outermost curve clearly demonstrates the trade-off conflicts at the frontier. Our experimental framework employs four reward models (CLIP, ICT, HPS, HP) based on different training data and optimization objectives, which naturally form a Pareto frontier in the reward distribution of generated images. This inherently demonstrates that our experiments have been conducted under conditions where reward conflicts exist.
>
>
>
> 2.  **Experimental Evidence:** To demonstrate the conflict between text-image alignment reward and human preference reward, we optimize Stable Diffusion 3.5 Turbo using CLIP Model and present the trend of CLIP score and human preference score throughout training in Table 1. The table shows that when optimizing solely for CLIP, the text-image alignment
> scores (CLIP/ICT) improve by +7.3%/+6.2%, while human preference scores (HPS/HP)
> degrade by -2.8%/-4.4%, clearly demonstrating the inherent reward conflicts.
>
>
>
>     **Table 1: Image-Text Alignment and Human Preference Score Trends during CLIP-only Optimization.**
>
>     | Metric | Type | Base |  100 |  200 |  300 |  400 | Trend |
>     |--------|------|------|----------|----------|----------|----------|-------|
>     | CLIP | Image-Text Alignment | 0.2943 | 0.2944 | 0.3043 | 0.3136 | 0.3157 | ↑ |
>     | ICT | Image-Text Alignment | 0.7738 | 0.7738 | 0.7758 | 0.8082 | 0.8215 | ↑ |
>     | HPS | Human Preference | 0.2558 | 0.2558 | 0.2556 | 0.2492 | 0.2487 | ↓ |
>     | HP | Human Preference | 0.7789 | 0.7789 | 0.7785 | 0.7711 | 0.7448 | ↓ |
>
>
>
>
> **Q2:** Is this method applicable to arbitrary text-to-image post-training approaches?
>
> **A2:** Our method is broadly applicable to any gradient-based post-training approach. We designed our framework to be training-algorithm-agnostic, requiring only that the base method uses rewards to provide optimization signals. To demonstrate this generalizability, we conduct experiments on two representative post-training methods with different optimization strategies: DRAFT-K[1] (shown in main paper) and ReFL[2] (shown in Table 2).
>
>
> **Table 2: Quantitative Results (%) on Parti-Prompts Dataset using ReFL[2].**
>
> | Model | ICT (%)↑ | HP (%)↑ | CLIP (%)↑ | HPS (%)↑ | JDR₂ (%)↑ | JDR₄ (%)↑ | JCR₄ (%)↓ |
> |:------|:--------:|:-------:|:---------:|:--------:|:---------:|:---------:|:---------:|
> | ReFL (Baseline) | 51.05|	51.50 |**50.50**|64.51 |24.52 | 9.24 | 7.45|
> | ReFL (**Ours**) |  **52.08**| **61.46**| 49.69|**68.32** |**31.86** | **13.54**| **4.72**|
>
>
>
>
>
>
>
>
> **References**
>
> [1] Kevin Clark, Paul Vicol, Kevin Swersky, and David J. Fleet. Directly Fine-Tuning Diffusion Models on Differentiable Rewards. In The Twelfth International Conference on Learning Representations (ICLR 2024).
>
> [2] Jiazheng Xu, Xiao Liu, Yuchen Wu, Yuxuan Tong, Qinkai Li, Ming Ding, Jie Tang, and Yuxiao Dong. ImageReward: Learning and Evaluating Human Preferences for Text-to-Image Generation. In Advances in Neural Information Processing Systems 36 (NeurIPS 2023).

---

### Author Response · Authors · 2025-12-03
**Discussion Summary**

We sincerely thank the Area Chair for overseeing the review of our work, and all reviewers for their constructive feedback, which has substantially improved our paper.

During the rebuttal period, we provided detailed responses and comprehensive experiments addressing all concerns raised. Notably, before the discussion period was unexpectedly closed, Reviewer PiGz explicitly stated: "Thank you for the detailed and well-structured rebuttal. It has addressed my main concerns and clarified the issues I raised. I am satisfied with the authors' responses and will keep my positive rating for this paper." This demonstrates that our responses effectively resolved the reviewers' core concerns.

We have incorporated all relevant experimental results into the revised manuscript, including **9 new experiments and 3 new sections** that directly address the reviewers' questions. Below we summarize the key concerns and our responses.

### Key Concerns and Responses:

#### 1: Reward conflicts (Reviewer `Wabn`)
We have updated Section 3.1 to clarify that "conflict" in Pareto optimization refers to trade-offs at the frontier rather than global opposition. Experimental evidence in Appendix C (Table 6) demonstrates that optimizing solely for the CLIP model improves text-image alignment by +7.3%/+6.2% while degrading human preference by -2.8%/-4.4%, clearly establishing inherent conflicts between these reward objectives.

#### 2: GPT-4o agent reproducibility and scalability (Reviewers `Xf1N` and `PiGz`)
We have revised Section 4.3 to address these concerns comprehensively. Our results show that GPT-4o achieves 90.5% agreement with human experts, while open-source alternatives reach 87% accuracy (Table 5), demonstrating both reproducibility and accessibility. The agent incurs only 0.4% of total training overhead through periodic invocation. Our method scales with O(n) linear complexity compared to the O(m^n) combinatorial explosion of weighted-sum approaches. Furthermore, we demonstrate in Appendix D.4 (Tables 7-8) that simple heuristic methods (mean, variance, KL divergence) fail to reliably detect reward hacking, validating the necessity of our agent-based approach.

#### 3: Computational overhead and optimal transport benefits (Reviewers `aapt` and `PiGz`)
We have added a detailed computational analysis in Appendix G. Our method introduces minimal additional overhead: while the one-time 2-hour precomputation accounts for 12% of training time, it enables 5× faster convergence; the per-iteration overhead increases by only 3% (baseline 6.0s vs. ours 6.18s), while effectively preventing reward hacking that baselines cannot address. The ablation study in Appendix E (Table 10) demonstrates that our OT-based distribution-level optimization consistently outperforms weighted sum, separate constraints, and simple Pareto mapping approaches across all metrics.

### Other Small Issues Raised by Reviewers:

| **Category** | **Concern** | **New Section/Experiment** | **Revision Details** |
|--------------|-------------|----------------------------|----------------------|
| **Sensitivity** | Hyperparameter sensitivity (Reviewer `Xf1N`) | Appendix E, Table 11 | **New** Table 11 shows robust performance across entropy regularization values (ε = 0.1/0.5/0.9). |
| **Scalability** | Applicability to different post-training algorithms (Reviewer `Wabn`) | Appendix E, Table 12| **New** Table 12 validates effectiveness on ReFL, confirming broad applicability beyond DRAFT-K. |
|  | Scalability to larger reward sets (Reviewer `aapt`) | Appendix E, Table 13 | **New** Table 13 demonstrates strong performance with 6 rewards with only 8% overhead increase. |
| **Method Design** | A priori reward strength detection (Reviewer `Xf1N`) | Appendix D.5, Table 9 | **New** Table 9 shows strong rewards significantly outperform weak rewards in human preference prediction accuracy (87.58%/88.47% vs. 60.30%/72.88%). |
| | Online Pareto frontier construction (Reviewer `Xf1N`) | Section 3.4 | **Updated** Section 3.4 provides step-by-step explanation of how the online frontier is dynamically constructed from generated samples. |
| | Out-of-distribution generalization (Reviewer `Xf1N`) | Section 5 | **Clarified** All experiments validated on two OOD datasets: Parti-Prompts and DiffusionDB. |

We believe these revisions have substantially strengthened the paper and adequately addressed all reviewers' concerns. We sincerely appreciate the Area Chair's dedicated effort throughout this review process.

---

### Meta-Review · Area_Chair_nAnz · 2026-01-02

**Summary:**

This paper works on preference optimization for text-to-image generation. Specifically authors tried to mitigate reward hacking in multi-reward optimization for text-to-image generation. Authors proposed a Pareto frontier-guided optimal transport framework, creating a frontier for each prompt as the optimization target and maps generated samples within the same batch to their corresponding frontiers. Authors proposed both online and offline optimization strategies to adopt to different reward models. They also introduced JDR and JCR to evaluate multi-reward optimization. Experimental results show the effectiveness of the proposed methods.

Before rebuttal, this paper got three 4 ratings and one 6 rating.

The strength of this paper given by reviewers are:
1. "heterogeneous reward bounds vs. a unified global target" is interesting and providing a theoretical foundation. (reviewer Xf1N, aapt, Wabn, PiGz)
2. Pareto frontiers and Optimal Transport is a natural and mathematically sound approach for multi-objective optimization. (reviewer Xf1N, aapt, PiGz)
3. evaluation is comprehensive. (reviewer Xf1N)
4. Joint Dominance Ratio (JDR) and Joint Collapse Ratio (JCR) provide a more precise way to evaluate the effectiveness of different training strategies. (reviewer Xf1N, PiGz)
5. paper is well written and easy to follow. (reviewer Wabn)

The weakness & questions of this paper given by reviewers are:
1. use of GPT-4o as a core component in the training loop is a potential weakness. (reviewer Xf1N, PiGz)
2. The link between the formal definition and the practical implementation could be tighter. (reviewer Xf1N)
3. sensitivity of hyperparameters. (reviewer Xf1N)
4. overhead for offline precomputation. (reviewer Xf1N, aapt)
5. need clarification of some design details. (reviewer Xf1N, aapt)
6. generalizes to other text-to-image models or scales with an increased number of reward models or other methods? (reviewer aapt, Wabn)
7. robustness of classification of reward models. (reviewer aapt)
8. more representative rewards selection? (reviewer Wabn)
9. need more comparison with other methods like non-dominated sorting or a simpler projection to the frontier. (reviewer PiGz)

In the discussion, reviewer PiGz suggested authors addressed their concern and kept their rating.

AC read the paper, reviewers' comments, rebuttal carefully, and found that although authors addressed some of the reviewers' concern, some critical concerns like diversified reward selection, clear definition of "human-perceived quality", why GPT-4o is a good proxy, some important details like how human expert evaluation make decision on when and which reward caused reward hacking are either  unaddressed or missing. Details are in the below session. Given these concerns or confusions it is hard for AC to make a positive decision and have to reject this paper. Hope authors will find reviewers' comments are useful for their future research.

**Reviewer Concerns:**

weakness 1. reviewer PiGz explicitly mentioned their concerns are addressed. for the questions "heuristic methods for detecting reward hacking." authors provided results show heuristic methods doesn't work, but they also didn't provide results show their methods works for this case. Also it is not clear how human expert evaluation make decision on when and which reward caused reward hacking.

weakness 2. this is not answered well. actually authors didn't give a clear definition of "human-perceived quality". There are many aspects for human-perceived quality for text to image generation images, for example, artifacts, image-text alignment, aesthetics. It is not clear what doesn't authors' mean by using "human-perceived quality". Given there are many research work on quality evaluation for text-to-image generation, there is definitely possibility there are better things than GPT-4o.

weakness 3. for the data authors provide, the final results do sensitive to hyperparameters.

weakness 4. authors provided numbers.

weakness 5. authors provided more details.

weakness 6. authors provided more results and addressed reviewer Wabn's concern. But missing results for other 2 rewards in table 2 when replied to reviewer aapt.

weakness 7. authors provide answers to address reviewer's concerns.

weakness 8. authors provided results shows that when optimizing solely for CLIP, the text-image alignment scores (CLIP/ICT) improve by +7.3%/+6.2%, while human preference scores (HPS/HP) degrade by -2.8%/-4.4%, clearly demonstrating the inherent reward conflicts. But AC think the paper still could benefit from more diversities rewards selection.

weakness 9. reviewer PiGz explicitly mentioned their concerns are addressed.

**Reviewer Scores:**

Reviewer Wabn will keep their rating 4.

Reviewer Xf1N will keep their rating 4.

Reviewer aapt might increase their rating from 4.

Reviewer PiGz said they kept their rating 6.

---

### Decision · Program_Chairs · 2026-01-26

Reject